# CALIBRATED DATASET CONDENSATION FOR FASTER HYPERPARAMETER SEARCH

## ABSTRACT

Dataset condensation can be used to reduce the computational cost of training multiple models on a large dataset by condensing the training dataset into a small synthetic set. State-of-the-art approaches rely on matching model gradients between the real and synthetic data. However, there is no theoretical guarantee on the generalizability of the condensed data: data condensation often generalizes poorly *across hyperparameters/architectures* in practice. In this paper, we consider a different condensation objective specifically geared toward *hyperparameter search*. We aim to generate a synthetic validation dataset so that the validation-performance rankings of models, with different hyperparameters, on the condensed and original datasets are comparable. We propose a novel *hyperparameter-calibrated dataset condensation* (HCDC) algorithm, which obtains the synthetic validation dataset by matching the *hyperparameter gradients* computed via implicit differentiation and efficient inverse Hessian approximation. Experiments demonstrate that the proposed framework effectively maintains the validation-performance rankings of models and speeds up hyperparameter/architecture search for tasks on both images and graphs.

## 1 INTRODUCTION

Deep learning has achieved great success in various fields, such as computer vision and graph related tasks. However, the computational cost of training state-of-the-art neural networks is rapidly increasing due to growing model and dataset sizes. Moreover, designing deep learning models usually requires training numerous models on the same data to obtain the optimal hyperparameters and architecture (Elsken et al., 2019), posing significant computational challenges. Thus, reducing the computational cost of repeatedly training on the same dataset is crucial. We address this problem from a data-efficiency perspective and consider the following question: how can one reduce the training data size for faster *hyperparameter search/optimization* with minimal performance loss?

Recently, *dataset distillation/condensation* (Wang et al., 2018) is proposed as an effective way to reduce sample size. This approach involves producing a small *synthetic* dataset to replace the original larger one, so that the test performance of the model trained on the synthetic set is comparable to that trained on the original. Despite the state-of-the-art performance achieved by recent dataset condensation methods when used to train a single pre-specified model, it remains challenging to utilize such methods effectively for hyperparameter search. Current dataset condensation methods perform poorly when applied to neural architecture search (NAS) (Elsken et al., 2019) and when used to train deep networks beyond the pre-specified architecture (Cui et al., 2022). Moreover, there is little or even a negative correlation between the performance of models trained on the synthetic vs. the full dataset, across architectures: often, one architecture achieves higher validation accuracy when trained on the original data relative to a second architecture, but obtains lower validation accuracy than the second when trained on the synthetic data. Since architecture performance ranking is not preserved when the original data is condensed, current data condensation methods are inadequate for NAS. This issue stems from the fact that existing condensation methods are designed on top of a single pre-specified model, and thus the condensed data may overfit this model.

We ask: *is it possible to preserve the architecture/hyperparameter search outcome when the original data is replaced by the condensed data?*

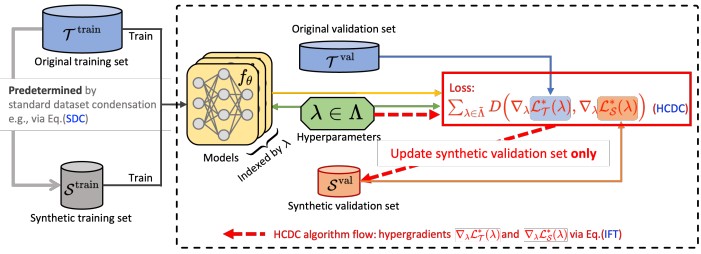

Figure 1: Hyperparameter Calibrated Dataset Condensation (HCDC) aims to find a small validation dataset such that the validation-performance rankings of the models with different hyperparameters are comparable to the large original dataset's. Our method realizes this goal $\big($Eq. (HC)$\big)$ by learning the synthetic validation set to match the hypergradients w.r.t the hyperparameters $\big($Eq. (HCDC) in the "Loss" box$\big)$. *Our contribution is depicted within the big black dashed box: the algorithm flow is indicated through the red dashed arrows.* The synthetic training set is predetermined by any standard dataset condensation (SDC) methods (e.g., Eq. (SDC)). The synthetic training and validation datasets obtained can later be used for hyperparameter search using only a fraction of the original computational load. A more detailed diagram is depicted in Fig. 5 in Appendix A.

To answer this question, we reformulate the dataset condensation problem using a hyperparameter optimization (HPO) framework (Feurer & Hutter, 2019), with the goal of preserving architecture/hyperparameter search outcomes over *multiple* architectures/hyperparameters, just as standard dataset condensation preserves generalization performance results for a *single pre-specified* architecture. This is illustrated in Fig. 1's "Goal" box. However, solving the resulting nested optimization problem is tremendously difficult. Therefore, we consider an alternative objective and show that architecture performance ranking preservation is equivalent to aligning the *hyperparameter gradients* (or *hypergradients* for short), of this objective, in the context of dataset condensation. This is illustrated as the "Loss" box in Fig. 1. Thus, we propose *hyperparameter calibrated dataset condensation (HCDC)*, a novel condensation method that preserves hyperparameter performance rankings by aligning the hypergradients computed using the condensed data to those computed using the original dataset, see Fig. 1.

Our implementation of HCDC is efficient and scales linearly with respect to the size of hyperparameter search space. Moreover, hypergradients are efficiently computed with constant memory overhead, using the implicit function theorem (IFT) and the Neumann series approximation of an inverse Hessian (Lorraine et al., 2020). We also specifically consider how to apply HCDC to the practical architecture search spaces for image and graph datasets.

Experiments demonstrate that our proposed HCDC algorithm drastically increases the correlation between the architecture rankings of models trained on the condensed dataset and those trained on the original dataset, for both image and graph data. Additionally, the test performance of the highest ranked architecture determined by the condensed dataset is comparable to that of the true optimal architecture determined by the original dataset. Thus, HCDC can enable faster hyperparameter search and obtain high performance accuracy by choosing the highest ranked hyperparameters, while the other condensation and coreset methods cannot. We also demonstrate that condensed datasets obtained with HCDC are compatible with off-the-shelf architecture search algorithms with or without parameter sharing.

We summarize our contributions as follows: **(1)** We study the data condensation problem for hyperparameter search and show that performance ranking preservation is equivalent to hypergradient alignment in this context. **(2)** We propose HCDC, which synthesizes condensed data by aligning the hypergradients of the objectives associated with the condensed and original datasets for faster hyperparameter search. **(3)** We present experiments, for which HCDC drastically reduces the search time and complexity of off-the-shelf NAS algorithms, for both image and graph data, while preserving the search outcome with high accuracy.

## 2 STANDARD DATASET CONDENSATION

Consider a classification problem where the original dataset $\mathcal{T}^{\text{train}} = \{(x_i, y_i)\}_{i=1}^{n}$ consists of $n$ (input, label) pairs sampled from the original data distribution $P_{\mathcal{D}}$. To simplify notation, we replace $\mathcal{T}^{\text{train}}$ with $\mathcal{T}$ when the context is clear. The classification task goal is to train a function $f_\theta$ (e.g., a deep neural network), with parameter $\theta$, to correctly predict labels $y$ from inputs $x$. Obtaining $f_\theta$

involves optimizing an empirical loss objective determined by $\mathcal{T}^{\text{train}}$:

$$\theta^{\mathcal{T}} = \arg\min_{\theta} \mathcal{L}_{\mathcal{T}}^{\text{train}}(\theta, \lambda), \text{ where } \mathcal{L}_{\mathcal{T}}^{\text{train}}(\theta, \lambda) \coloneqq \frac{1}{|\mathcal{T}^{\text{train}}|} \sum_{(x,y) \in \mathcal{T}^{\text{train}}} l(f_{\theta}(x), y, \lambda), \quad (1)$$

where $\lambda$ denotes the model hyperparameter (e.g., the neural network architecture that characterizes $f_{\theta}$), and $l(\cdot, \cdot, \cdot)$ is a task-specific loss function that depends on $\lambda$.

Dataset condensation involves generating a small set of $c \ll n$ synthesized samples $\mathcal{S} = \{x_i', y_i'\}_{i=1}^c$, with which to replace the original training dataset $\mathcal{T}$. Using the condensed dataset $\mathcal{S}$, one can obtain $f_{\theta}$ with parameter $\theta = \theta^{\mathcal{S}} = \arg\min_{\theta} \mathcal{L}_{\mathcal{S}}^{\text{train}}(\theta, \lambda)$, where $\mathcal{L}_{\mathcal{S}}^{\text{train}} = \frac{1}{|\mathcal{S}|} \sum_{(x,y) \in \mathcal{S}} l(f_{\theta}(x), y, \lambda)$. The goal is for the generalization performance of the model $f_{\theta^{\mathcal{S}}}$ obtained using the condensed data to approximate that of $f_{\theta^{\mathcal{T}}}$, i.e., $\mathbb{E}_{(x,y) \sim P_{\mathcal{D}}}[l(f_{\theta^{\mathcal{T}}}(x), y, \lambda)] \approx \mathbb{E}_{(x,y) \sim P_{\mathcal{D}}}[l(f_{\theta^{\mathcal{S}}}(x), y, \lambda)]$.

Next, we review the bi-level optimization formulation of the *standard dataset condensation* (SDC) (Wang et al., 2018) and one of its efficient solutions using gradient matching (Zhao et al., 2020).

**SDC's objective.** By posing the optimal parameters $\theta^{\mathcal{S}}(\mathcal{S})$ as a function of the condensed dataset $\mathcal{S}$, SDC can be formulated as a bi-level optimization problem as follows,

$$\mathcal{S}^* = \arg\min_{\mathcal{S}} \mathcal{L}_{\mathcal{T}}^{\text{train}}(\theta^{\mathcal{S}}(\mathcal{S}), \lambda), \text{ s.t. } \theta^{\mathcal{S}}(\mathcal{S}) \coloneqq \arg\min_{\theta} \mathcal{L}_{\mathcal{S}}^{\text{train}}(\theta, \lambda). \quad \text{(SDC)}$$

In other words, the optimization problem in Eq. (SDC) aims to find the optimal synthetic dataset $\mathcal{S}$ such that the model $\theta^{\mathcal{S}}(\mathcal{S})$ trained on it minimizes the training loss over the original data $\mathcal{T}^{\text{train}}$. However, directly solving the optimization problem in Eq. (SDC) is difficult since it involves a nested-loop optimization and solving the inner loop for $\theta^{\mathcal{S}}(\mathcal{S})$ at each iteration requires unrolling the recursive computation graph for $\mathcal{S}$ over multiple optimization steps for $\theta$ (Domke, 2012), which is computationally expensive.

**SDC in a gradient matching formulation.** Zhao et al. (2020) alleviate this computational issue by introducing a *gradient matching* (GM) formulation. Firstly, they formulate the condensation objective as not only achieves comparable generalization performance to $\theta^{\mathcal{T}}$ but also converges to a similar solution in the parameter space, i.e., $\theta^{\mathcal{S}}(\mathcal{S}, \theta_0) \approx \theta_{\mathcal{T}}(\theta_0)$, where $\theta_0$ indicates the initialization. The resulting formulation is still a bilevel optimization but can be simplified via several approximations.

*(1)* $\theta^{\mathcal{S}}(\mathcal{S}, \theta_0)$ is approximated by the output of a series of gradient-descent updates, $\theta^{\mathcal{S}}(\mathcal{S}, \theta_0) \approx \theta_{t+1}^{\mathcal{S}} \leftarrow \theta_t^{\mathcal{S}} - \eta \nabla_{\theta} \mathcal{L}_{\mathcal{S}}^{\text{train}}(\theta_t^{\mathcal{S}}, \lambda)$. In addition, Zhao et al. (2020) propose to match $\theta_{t+1}^{\mathcal{S}}$ with incompletely optimized $\theta_{t+1}^{\mathcal{T}}$ at each iteration $t$. Consequently, the dataset condensation objective is now $\mathcal{S}^* = \arg\min_{\mathcal{S}} \mathbb{E}_{\theta_0 \sim P_{\theta_0}}[\sum_{t=0}^{T-1} D(\theta_t^{\mathcal{S}}, \theta_t^{\mathcal{T}})]$.

*(2)* If we assume $\theta_t^{\mathcal{S}}$ can always track $\theta_t^{\mathcal{T}}$ (i.e., $\theta_t^{\mathcal{S}} \approx \theta_t^{\mathcal{T}}$) from the initialization $\theta_0$ up to iteration $t$, then we can replace $D(\theta_{t+1}^{\mathcal{S}}, \theta_{t+1}^{\mathcal{T}})$ by $D(\nabla_{\theta} \mathcal{L}_{\mathcal{S}}^{\text{train}}(\theta_t^{\mathcal{S}}, \lambda), \nabla_{\theta} \mathcal{L}_{\mathcal{T}}^{\text{train}}(\theta_t^{\mathcal{T}}, \lambda))$. The final objective for the GM formulation is,

$$\min_{\mathcal{S}} \mathbb{E}_{\theta_0 \sim P_{\theta_0}} \Big[ \sum_{t=0}^{T-1} D\Big( \nabla_{\theta} \mathcal{L}_{\mathcal{S}}^{\text{train}}(\theta_t^{\mathcal{S}}, \lambda), \nabla_{\theta} \mathcal{L}_{\mathcal{T}}^{\text{train}}(\theta_t^{\mathcal{S}}, \lambda) \Big) \Big]. \quad (2)$$

**Challenge of varying hyperparameter $\lambda$.** In the formulation of the SDC, the condensed data $\mathcal{S}$ is learned with a fixed hyperparameter $\lambda$, e.g., a pre-specified neural network architecture. As a result, the condensed data trained with SDC's objective performs poorly on hyperparameter search (Cui et al., 2022), which requires the performance of models under varying hyperparameters to behave consistently on the original and condensed dataset. In the following, we tackle this issue by reformulating the dataset condensation problem under the hyperparameter optimization framework.

## 3 HYPERPARAMETER CALIBRATED DATASET CONDENSATION

In this section, we would like to develop a condensation method specifically for preserving the outcome of *hyperparameter optimization* (HPO) on the condensed dataset across different architectures/hyperparameters for faster hyperparameter search. This requires dealing with varying choices of hyperparameters so that the relative performances of different hyperparameters on the condensed and

original datasets are consistent. We first formulate the data condensation for hyperparameter search in the HPO framework below and then propose the hyperparameter calibrated dataset condensation framework in Section 4 by using the equivalence relationship between preserving the performance ranking and the hypergradient alignment.

**HPO's objective.** Given $\mathcal{T} = \mathcal{T}^{\text{train}} \bigcup \mathcal{T}^{\text{val}} \bigcup \mathcal{T}^{\text{test}}$, HPO aims to find the optimal hyperparameter $\lambda^{\mathcal{T}}$ that minimizes the validation loss of the model optimized on the training dataset $\mathcal{T}^{\text{train}}$ with hyperparameter $\lambda^{\mathcal{T}}$, i.e.,

$$\lambda^{\mathcal{T}} = \arg\min_{\lambda \in \Lambda} \mathcal{L}^*_{\mathcal{T}}(\lambda), \text{ where } \mathcal{L}^*_{\mathcal{T}}(\lambda) \coloneqq \mathcal{L}^{\text{val}}_{\mathcal{T}}(\theta^{\mathcal{T}}(\lambda), \lambda) \text{ and } \theta^{\mathcal{T}}(\lambda) \coloneqq \arg\min_{\theta} \mathcal{L}^{\text{train}}_{\mathcal{T}}(\theta, \lambda).$$
(HPO)

Here $\mathcal{L}^{\text{val}}_{\mathcal{T}}(\theta, \lambda) \coloneqq \frac{1}{|\mathcal{T}^{\text{val}}|} \sum_{(x,y) \in \mathcal{T}^{\text{val}}} l(f_\theta(x), y, \lambda)$. HPO is a bi-level optimization where both the optimal parameter $\theta^{\mathcal{T}}(\lambda)$ and the optimized validation loss $\mathcal{L}^*_{\mathcal{T}}(\lambda)$ are viewed as a function of the hyperparameter $\lambda$.

**Dataset condensation for HPO.** We would like to synthesize a condensed training dataset $\mathcal{S}^{\text{train}}$ and a condensed validation dataset $\mathcal{S}^{\text{val}}$ to replace the original $\mathcal{T}^{\text{train}}$ and $\mathcal{T}^{\text{val}}$ for hyperparameter search. Denote the synthetic dataset as $\mathcal{S} = \mathcal{S}^{\text{train}} \bigcup \mathcal{S}^{\text{val}}$. Similar to Eq. (HPO), the optimal hyperparameter $\lambda^{\mathcal{S}}$ is defined for a given dataset $\mathcal{S}$. Naively, one can formulate such a problem as finding the condensed dataset $\mathcal{S}$ to minimize the validation loss on the original dataset $\mathcal{T}^{\text{val}}$ as follows, which is an optimization problem similar to the standard dataset condensation in Eq. (SDC):

$$\mathcal{S}^* = \arg\min_{\mathcal{S}} \mathcal{L}^*_{\mathcal{T}}\big(\lambda^{\mathcal{S}}(\mathcal{S})\big) \quad \text{s.t.} \quad \lambda^{\mathcal{S}}(\mathcal{S}) \coloneqq \arg\min_{\lambda \in \Lambda} \mathcal{L}^*_{\mathcal{S}}(\lambda),$$
(3)

where the optimized validation losses $\mathcal{L}^*_{\mathcal{T}}(\cdot)$ and $\mathcal{L}^*_{\mathcal{S}}(\cdot)$ are defined following Eq. (HPO).

However, **two challenges** exist for such a formulation. **Challenge (1)**: Eq. (3) is a nested optimization (for dataset condensation) over another nested optimization (for HPO), which is computationally expensive. **Challenge (2)**: the search space $\Lambda$ of the hyperparameters can be complicated. In contrast to parameter optimization, where the search space is usually assumed to be the continuous and unbounded Euclidean space, the search space of the hyperparameters can be compositions of discrete and continuous spaces. Having such discrete components in the search space poses challenges for gradient-based optimization methods.

To address **Challenge (1)**, we propose an alternative objective based on the alignment of hypergradients that can be computed efficiently in Section 4. For **Challenge (2)**, we construct the extended search space in Section 5.

## 4 HYPERPARAMETER CALIBRATION VIA HYPERGRADIENT ALIGNMENT

In this section, we introduce Hyperparameter-Calibrated Dataset Condensation (HCDC), a novel condensation method designed to align *hyperparameter gradients* – referred to as *hypergradients* – thus preserving the validation performance ranking of various hyperparameters.

**Hyperparameter calibration.** To tackle the computational challenges inherent in hyperparameter optimization (HPO) as expressed in Eq. (3), we propose an efficient yet sufficient alternative. Rather than directly solving the HPO problem, we aim to identify a condensed dataset that maintains the outcomes of HPO on the hyperparameter set $\Lambda$. We refer to this process as hyperparameter calibration, formally defined as follows.

**Definition 1** (Hyperparameter Calibration). *Given original dataset $\mathcal{T}$, generic model $f^\lambda_\theta$, and hyperparameter search space $\Lambda$, we say a condensed dataset $\mathcal{S}$ is hyperparameter calibrated, if for any $\lambda_1 \neq \lambda_2 \in \Lambda$, it holds that,*

$$\Big(\mathcal{L}^*_{\mathcal{T}}(\lambda_1) - \mathcal{L}^*_{\mathcal{T}}(\lambda_2)\Big)\Big(\mathcal{L}^*_{\mathcal{S}}(\lambda_1) - \mathcal{L}^*_{\mathcal{S}}(\lambda_2)\Big) > 0$$
(HC)

*In other words, changes of the optimized validation loss on $\mathcal{T}$ and $\mathcal{S}$ always have the same sign, between any pairs of hyperparameters $\lambda_1 \neq \lambda_2$.*

It is evident that if hyperparameter calibration (HC) is satisfied, the outcomes of HPO on both the original and condensed datasets will be identical. Consequently, our objective shifts to *ensuring hyperparameter calibration across all pairs of hyperparameters*.

**HCDC: hypergradient alignment objective for dataset condensation.** To move forward, we make the assumption that there exists a continuous extension of the search space. Specifically, the (potentially discrete) search space $\Lambda$ can be extended to a compact and connected set $\tilde{\Lambda} \supset \Lambda$. Within this extended set, we define a continuation of the generic model $f_\theta^\lambda$ such that $f_\theta^\lambda$ is differentiable anywhere in $\tilde{\Lambda}$. In Section 5, we will elaborate on how to construct such an extended search space $\tilde{\Lambda}$.

To establish a new objective for hyperparameter calibration, consider the case when $\lambda_1$ is in the neighborhood of $\lambda_2$, denoted as $\lambda_1 \in B_r(\lambda_2)$ for some $r > 0$. In this situation, the change in validation loss can be approximated up to first-order by the hypergradients, as follows: $\mathcal{L}_\mathcal{T}^*(\lambda_1) - \mathcal{L}_\mathcal{T}^*(\lambda_2) \approx \langle \nabla_\lambda \mathcal{L}_\mathcal{T}^*(\lambda_2), \Delta\lambda \rangle$. Here, $\Delta\lambda = \lambda_1 - \lambda_2$ with $r \geq \|\Delta\lambda\|_2 \to 0^+$. Analogously, for the synthetic dataset we have: $\mathcal{L}_\mathcal{S}^*(\lambda_1) - \mathcal{L}_\mathcal{S}^*(\lambda_2) \approx \langle \nabla_\lambda \mathcal{L}_\mathcal{S}^*(\lambda_2), \Delta\lambda \rangle$. Hence, the hyperparameter calibration condition simplifies to $\langle \nabla_\lambda \mathcal{L}_\mathcal{T}^*(\lambda_2), \Delta\lambda \rangle \cdot \langle \nabla_\lambda \mathcal{L}_\mathcal{S}^*(\lambda_2), \Delta\lambda \rangle > 0$. Further simplification leads to $\nabla_\lambda \mathcal{L}_\mathcal{T}^*(\lambda) \parallel \nabla_\lambda \mathcal{L}_\mathcal{S}^*(\lambda)$, indicating *alignment* of the two hypergradient vectors. We formally define this hypergradient alignment and establish its equivalence to hyperparameter calibration.

**Definition 2** (Hypergradient Alignment). *We say hypergradients are aligned in an extended search space $\tilde{\Lambda}$, if for any $\lambda \in \tilde{\Lambda}$, it holds that $\nabla_\lambda \mathcal{L}_\mathcal{T}^*(\lambda) \parallel \nabla_\lambda \mathcal{L}_\mathcal{S}^*(\lambda)$, i.e., $D_c(\nabla_\lambda \mathcal{L}_\mathcal{T}^*(\lambda), \nabla_\lambda \mathcal{L}_\mathcal{S}^*(\lambda)) = 0$, where $D_c(\cdot, \cdot) = 1 - \cos(\cdot, \cdot)$ represents the cosine distance.*

**Theorem 1** (**Equivalence between Hypergradient Alignment and Hyperparameter Calibration**).
*Hypergradient alignment (Definition 2) is equivalent to hyperparameter calibration (Definition 1) on a connected and compact set, e.g., the extended search space $\tilde{\Lambda}$.*

The implication is straightforward: if hyperparameter calibration holds in $\tilde{\Lambda}$, it also holds in $\Lambda$. According to Theorem 1, achieving hypergradient alignment in $\tilde{\Lambda}$ is sufficient to ensure hyperparameter calibration in $\Lambda$. Therefore, the integrity of the HPO outcome over $\Lambda$ is maintained.

Consequently, the essence of our hyperparameter calibrated dataset condensation (HCDC) is to align/match the hypergradients calculated on both the original and condensed datasets within the extended search space $\tilde{\Lambda}$:

$$\mathcal{S}^* = \arg\min_\mathcal{S} \sum_{\lambda \in \tilde{\Lambda}} D_c\Big( \nabla_\lambda \mathcal{L}_\mathcal{T}^{\mathrm{val}}\big(\theta^\mathcal{T}(\lambda), \lambda\big), \nabla_\lambda \mathcal{L}_\mathcal{S}^{\mathrm{val}}\big(\theta^\mathcal{S}(\lambda), \lambda\big) \Big), \qquad \text{(HCDC)}$$

where the cosine distance $D_c(\cdot, \cdot) = 1 - \cos(\cdot, \cdot)$ is used without loss of generality.

## 5 IMPLEMENTATIONS OF HCDC

In this section, we focus on implementing the hyperparameter calibrated dataset condensation (HCDC) algorithm. We address two primary challenges: **(1)** efficient approximate computation of hyperparameter gradients, often called hypergradients, using implicit differentiation techniques; and **(2)** the efficient formation of the extended search space $\tilde{\Lambda}$. The complete pseudocode for HCDC will be provided at the end of this section.

### 5.1 EFFICIENT EVALUATION OF HYPERGRADIENTS

The efficient computation of hypergradients is well-addressed in existing literature (see Section 6). In our HCDC implementation, we utilize the implicit function theorem (IFT) and the Neumann series approximation for inverse Hessians, as proposed by Lorraine et al. (2020).

**Computing hypergradients via IFT.** The hypergradients are the gradients of the optimized validation loss $\mathcal{L}_\mathcal{T}^*(\lambda) = \mathcal{L}_\mathcal{T}^{\mathrm{val}}(\theta^\mathcal{T}(\lambda), \lambda)$ with respect to the hyperparameters $\lambda$; see Appendix E for further details. The implicit function theorem (IFT) provides an efficient approximation to compute the hypergradients $\nabla_\lambda \mathcal{L}_\mathcal{T}^*(\lambda)$ and $\nabla_\lambda \mathcal{L}_\mathcal{S}^*(\lambda)$.

$$\nabla_\lambda \mathcal{L}_\mathcal{T}^*(\lambda) \approx -\Big[ \frac{\partial^2 \mathcal{L}_\mathcal{T}^{\mathrm{train}}(\theta, \lambda)}{\partial\lambda \partial\theta^T} \Big] \Big[ \frac{\partial^2 \mathcal{L}_\mathcal{T}^{\mathrm{train}}(\theta, \lambda)}{\partial\theta \partial\theta^T} \Big]^{-1} \nabla_\theta \mathcal{L}_\mathcal{T}^{\mathrm{val}}(\theta, \lambda) + \nabla_\lambda \mathcal{L}_\mathcal{T}^{\mathrm{val}}(\theta, \lambda), \qquad \text{(IFT)}$$

where we consider the direct gradient $\nabla_\lambda \mathcal{L}_\mathcal{T}^{\mathrm{val}}(\theta, \lambda)$ is 0, since in most cases the hyperparameter $\lambda$ only affects the validation loss $\mathcal{L}_\mathcal{T}^{\mathrm{val}}(\theta, \lambda)$ through the model function $f_{\theta, \lambda}$. The first term consists of the mixed partials $\big[ \frac{\partial^2 \mathcal{L}_\mathcal{T}^{\mathrm{train}}(\theta, \lambda)}{\partial\lambda \partial\theta^T} \big]$, the inverse Hessian $\big[ \frac{\partial^2 \mathcal{L}_\mathcal{T}^{\mathrm{train}}(\theta, \lambda)}{\partial\theta \partial\theta^T} \big]^{-1}$, and the validation gradients $\nabla_\theta \mathcal{L}_\mathcal{T}^{\mathrm{val}}(\theta, \lambda)$. While the other parts can be calculated efficiently through a single back-propagation,

approximating the inverse Hessian is required. Lorraine et al. (2020) propose a stable, tractable, and efficient Neumann series approximation of the inverse Hessian as follows:

$$\left[\frac{\partial^2 \mathcal{L}_{\mathcal{T}}^{\text{train}}(\theta,\lambda)}{\partial\theta\partial\theta^T}\right]^{-1} = \lim_{i\to\infty} \sum_{j=0}^{i} \left[I - \frac{\partial^2 \mathcal{L}_{\mathcal{T}}^{\text{train}}(\theta,\lambda)}{\partial\theta\partial\theta^T}\right]^j,$$

which requires only constant memory. When combined with Eq. (IFT), the approximated hypergradients can be evaluated by employing efficient vector-Jacobian products (Lorraine et al., 2020).

**Optimizing hypergradient alignment loss in Eq. (HCDC).** To optimize the objective defined in HCDC (Eq. (HCDC)), we learn the synthetic validation set $\mathcal{S}^{\text{val}}$ from scratch. This is crucial as the hypergradients with respect to the validation losses in Eq. (HCDC), are significantly influenced by the synthetic validation examples, which are free learnable parameters during the condensation process. In contrast, we maintain the synthetic training set $\mathcal{S}^{\text{train}}$ as fixed. For generating $\mathcal{S}^{\text{train}}$, we employ the standard dataset condensation (SDC) algorithm, as described in Eq. (2). To optimize the synthetic validation set $\mathcal{S}^{\text{val}}$ with respect to the hyper-gradient loss in Eq. (HCDC), we compute the gradients of $\nabla_\theta \mathcal{L}_{\mathcal{S}}^{\text{val}}(\theta,\lambda)$ and $\nabla_\lambda \mathcal{L}_{\mathcal{S}}^{\text{val}}(\theta,\lambda)$ w.r.t. $\mathcal{S}^{\text{val}}$. This is handled using an additional back-propagation step, akin to the one in SDC that calculates the gradients of $\nabla_\theta \mathcal{L}_{\mathcal{S}}^{\text{train}}(\theta,\lambda)$ w.r.t $\mathcal{S}^{\text{train}}$.

## 5.2 Efficient Design of Extended Search Space

HCDC's objective (Eq. (HCDC)) necessitates the alignment of hypergradients across all hyperparameters $\lambda$'s in an extended space $\tilde{\Lambda}$. This space is a compact and connected superset of the original search space $\Lambda$. For practical implementation, we evaluate the hypergradient matching loss using a subset of $\lambda$ values randomly sampled from $\tilde{\Lambda}$. To enhance HCDC's efficiency within a predefined search space $\lambda$, our goal is to minimally extend this space to $\tilde{\Lambda}$ for sampling.

In the case of continuous hyperparameters, $\Lambda$ is generally both compact and connected, rending $\tilde{\Lambda}$ identical to $\Lambda$. For discrete search spaces $\Lambda$ consisting of $p$ candidate hyperparameters, we propose a linear-complexity construction for $\tilde{\Lambda}$ (in which the linearity is in terms of $p$). Specifically, for each $i \in [p]$, we formulate an "$i$-th HPO trajectory", a representative path that originates from $\lambda_{i,0}^{\mathcal{S}} = \lambda_i \in \Lambda$ and evolves via the update rule $\lambda_{i,t+1}^{\mathcal{S}} \leftarrow \lambda_{i,t}^{\mathcal{S}} - \eta \nabla_\lambda \mathcal{L}_{\mathcal{S}}^{\mathcal{S}}(\lambda_{i,t}^{\mathcal{S}})$, see Appendix H for details and Fig. 8 for illustration. We assume that all $p$ trajectories converge to the same or equivalent optima $\lambda^{\mathcal{S}}$, thus forming "connected" paths. Consequently, the extended search space $\tilde{\Lambda}$ comprises these $p$ connected trajectories, allowing us to evaluate the hypergradient matching loss along each trajectory $\{\lambda_{i,t}^{\mathcal{S}}\}_{t=0}^{T}$ during the iterative update of $\lambda$.

## 5.3 Pseudocode

We conclude this section by outlining the HCDC algorithm in Algorithm 1, assuming a discrete and finite hyperparameter search space $\Lambda$. In Line 7, we calculate the hypergradients $\nabla_\lambda \mathcal{L}_{\mathcal{S}}^*(\lambda)$ using Eq. (IFT). For computing the gradient $\nabla_{\mathcal{S}^{\text{val}}} D\big(\nabla_\lambda \mathcal{L}_{\mathcal{T}}^*(\lambda), \nabla_\lambda \mathcal{L}_{\mathcal{S}}^*(\lambda)\big)$ in Line 8, we note that only $\nabla_\lambda \mathcal{L}_{\mathcal{S}}^*(\lambda)$ is depends on $\mathcal{S}^{\text{val}}$. Employing Eq. (IFT), we find that $\nabla_\lambda \mathcal{L}_{\mathcal{S}}^*(\lambda) = -\left[\frac{\partial^2 \mathcal{L}_{\mathcal{S}}^{\text{train}}(\theta,\lambda)}{\partial\lambda\partial\theta^{\mathcal{S}}}\right]\left[\frac{\partial^2 \mathcal{L}_{\mathcal{S}}^{\text{train}}(\theta,\lambda)}{\partial\theta\partial\theta^{\mathcal{S}}}\right]^{-1} \nabla_\theta \mathcal{L}_{\mathcal{S}}^{\text{val}}(\theta,\lambda)$. Note that there are no direct gradients, as $\lambda$ influences the loss solely through the model $f_\theta^\lambda$. Therefore, to obtain $\nabla_{\mathcal{S}^{\text{val}}} D\big(\nabla_\lambda \mathcal{L}_{\mathcal{T}}^*(\lambda), \nabla_\lambda \mathcal{L}_{\mathcal{S}}^*(\lambda)\big)$, we simply need to compute the gradient $\nabla_{\mathcal{S}^{\text{val}}} \nabla_\theta \mathcal{L}_{\mathcal{S}}^{\text{val}}(\theta,\lambda)$ through standard back-propagation methods, since only the validation loss term $\nabla_\theta \mathcal{L}_{\mathcal{S}}^{\text{val}}(\theta,\lambda)$ depends on $\mathcal{S}^{\text{val}}$.

---

**Algorithm 1:** Hyperparameter Calibrated Dataset Condensation (HCDC)

**Input:** Original dataset $\mathcal{T}$, a set of NN architectures $f_\theta$, hyperparameter search space $\lambda \in \Lambda = \{\lambda_1, \ldots, \lambda_p\}$, predetermined condensed training data $\mathcal{S}^{\text{train}}$ learned by standard dataset condensation (e.g., Eq. (2)), randomly initialized synthetic examples $\mathcal{S}^{\text{val}}$ of $C$ classes.

1 **for** *repeat* $k = 0, \ldots, K - 1$ **do**
2   **foreach** *hyperparameters* $\lambda = \lambda_1, \ldots, \lambda_p$ *in* $\Lambda$ **do**
3     Initialize model parameters $\theta \leftarrow \theta_0 \sim P_{\theta_0}$
4     **for** *epoch* $t = 0, \ldots, T_\theta - 1$ **do**
5       Update model parameters $\theta \leftarrow \theta - \eta_\theta \nabla_\theta \mathcal{L}_{\mathcal{S}}^{\text{train}}(\theta,\lambda)$
6       **if** $t \mod T_\lambda = 0$ **then**
7         Update hyperparameters $\lambda \leftarrow \lambda - \eta_\lambda \nabla_\lambda \mathcal{L}_{\mathcal{S}}^*(\lambda)$
8       Update the synthetic validation set $\mathcal{S}^{\text{val}} \leftarrow \mathcal{S}^{\text{val}} - \eta_{\mathcal{S}} \nabla_{\mathcal{S}^{\text{val}}} D\big(\nabla_\lambda \mathcal{L}_{\mathcal{T}}^*(\lambda), \nabla_\lambda \mathcal{L}_{\mathcal{S}}^*(\lambda)\big)$
9 **return** *Condensed validation set* $\mathcal{S}^{\text{val}}$.

---

For a detailed complexity analysis of Algorithm 1 and further discussions, refer to Appendix I.

## 6 RELATED WORK

The traditional way to simplify a dataset is **coreset selection** (Toneva et al., 2018; Paul et al., 2021), where critical training data samples are chosen based on heuristics like diversity (Aljundi et al., 2019), distance to the dataset cluster centers (Rebuffi et al., 2017; Chen et al., 2010) and forgetfulness (Toneva et al., 2018). However, the performance of coreset selection methods is limited by the assumption of the existence of representative samples in the original data, which may not hold in practice.

To overcome this limitation, **dataset distillation/condensation** (Wang et al., 2018) has been proposed as a more effective way to reduce sample size. Dataset condensation (or dataset distillation) is first proposed in (Wang et al., 2018) as a learning-to-learn problem by formulating the network parameters as a function of synthetic data and learning them through the network parameters to minimize the training loss over the original data. This approach involves producing a small *synthetic* dataset to replace the original larger one, so that the test/generalization performance of the model trained on the synthetic set is comparable to that trained on the original. However, the nested-loop optimization precludes it from scaling up to large-scale in-the-wild datasets. Zhao et al. (2020) alleviate this issue by enforcing the gradients of the synthetic samples w.r.t. the network weights to approach those of the original data, which successfully alleviates the expensive unrolling of the computational graph. Based on the meta-learning formulation in (Wang et al., 2018), Bohdal et al. (2020) and Nguyen et al. (2020; 2021) propose to simplify the inner-loop optimization of a classification model by training with ridge regression which has a closed-form solution, while Such et al. (2020) model the synthetic data using a generative network. To improve the data efficiency of synthetic samples in the gradient-matching algorithm, Zhao & Bilen (2021a) apply differentiable Siamese augmentation, and Kim et al. (2022) introduce efficient synthetic-data parametrization.

**Implicit differentiation** methods apply the implicit function theorem (IFT) (Eq. (IFT)) to nested-optimization problems (Wang et al., 2019). Lorraine et al. (2020) approximated the inverse Hessian by Neumann series, which is a stable alternative to conjugate gradients (Shaban et al., 2019) and scales IFT to large networks with constant memory.

**Differentiable NAS** methods, e.g., DARTS (Liu et al., 2018) explore the possibility of transforming the discrete neural architecture space into a continuously differentiable form and further uses gradient optimization to search the neural architecture. SNAS (Xie et al., 2018) points out that DARTS suffers from the unbounded bias issue towards its objective, and it remodels the NAS and leverages the Gumbel-softmax trick (Jang et al., 2017; Maddison et al., 2017) to learn the architecture parameter.

In addition, we summarize more dataset condensation and coreset selection methods as well as graph reduction methods in Appendix B.

Table 1: The Spearman's rank correlation of architecture's performance (Corr.) and the test performance of the best architecture selected on the condensed dataset (Perf.) on two **image datasets**. Grid search is applied to find the best architecture.

| Dataset | | Coresets | | | Standard Condensation | | | | | Ours | Oracle |
|---|---|---|---|---|---|---|---|---|---|---|---|
| | | Random | K-Center | Herding | DC | DSA | DM | KIP | TM | HCDC | Optimal |
| CIFAR-10 | Corr. | $-0.12 \pm 0.07$ | $0.19 \pm 0.12$ | $-0.05 \pm 0.08$ | $-0.21 \pm 0.15$ | $-0.33 \pm 0.09$ | $-0.10 \pm 0.15$ | $-0.27 \pm 0.15$ | $-0.07 \pm 0.04$ | $\mathbf{0.74 \pm 0.21}$ | — |
| | Perf. (%) | $91.3 \pm 0.2$ | $91.4 \pm 0.3$ | $90.2 \pm 0.9$ | $89.2 \pm 3.3$ | $73.5 \pm 7.2$ | $92.2 \pm 0.4$ | $91.8 \pm 0.2$ | $75.2 \pm 4.3$ | $\mathbf{92.9 \pm 0.7}$ | 93.5 |
| CIFAR-100 | Corr. | $-0.05 \pm 0.03$ | $-0.07 \pm 0.05$ | $0.08 \pm 0.11$ | $-0.13 \pm 0.02$ | $-0.28 \pm 0.05$ | $-0.15 \pm 0.07$ | $-0.08 \pm 0.04$ | $-0.09 \pm 0.03$ | $\mathbf{0.63 \pm 0.13}$ | — |
| | Perf. (%) | $71.1 \pm 1.4$ | $69.5 \pm 2.8$ | $67.9 \pm 1.8$ | $64.9 \pm 2.2$ | $59.0 \pm 4.1$ | $70.1 \pm 0.6$ | $68.8 \pm 0.6$ | $51.3 \pm 6.1$ | $\mathbf{72.4 \pm 1.7}$ | 72.9 |

## 7 EXPERIMENTS

In this section, we validate the effectiveness of hyperparameter calibrated dataset condensation (HCDC) when applied to speed up architecture/hyperparameter search on two types of data: **images** and **graphs**. For an ordered list of architectures, we calculate Spearman's rank correlation coefficient $-1 \leq$ Corr. $\leq 1$, between the rankings of their validation performance on the original and condensed datasets. This correlation coefficient (denoted by **Corr.**) indicates how similar the performance ranking on the condensed dataset is to that on the original dataset. We also report the test accuracy (referred to as **Perf.**) evaluated on the original dataset of the architectures selected on the condensed dataset. If the test performance is close to the true optimal performance among all architectures, we say the architecture search outcome is preserved with high accuracy. See Appendix G and Appendix J for more discussions on implementation and experimental setups.

Table 2: Spearman's rank correlation of convolution filters in GNNs (Corr.) and the test performance of the best convolution filter selected on the condensed graph (Pref.) on four **graph datasets**. Continuous hyperparameter optimization (Lorraine et al., 2020) is applied to find the best convolution filter, while Spearman's rank correlation coefficients are evaluated on 80 sampled hyperparameter configurations. $n$ is the total number of nodes in the original graph, and $c_{train}$ is the number of training nodes in the condensed graph.

| Dataset | Ratio ($c_{train}/n$) | Random | | GCond-X | | GCond | | HCDC | | Whole Graph |
|---|---|---|---|---|---|---|---|---|---|---|
| | | Corr. | Perf. (%) | Corr. | Perf. (%) | Corr. | Perf. (%) | Corr. | Perf. (%) | Perf. (%) |
| Cora | 0.9% | $0.29 \pm .08$ | $81.2 \pm 1.1$ | $0.16 \pm .07$ | $79.5 \pm 0.7$ | $0.61 \pm .03$ | $81.9 \pm 1.6$ | $\mathbf{0.80 \pm .03}$ | $\mathbf{83.0 \pm 0.2}$ | |
| | 1.8% | $0.40 \pm .04$ | $81.9 \pm 0.5$ | $0.21 \pm .07$ | $80.3 \pm 0.4$ | $0.76 \pm .06$ | $83.2 \pm 0.9$ | $\mathbf{0.85 \pm .03}$ | $\mathbf{83.4 \pm 0.2}$ | $83.8 \pm 0.4$ |
| | 3.6% | $0.51 \pm .04$ | $82.2 \pm 0.6$ | $0.23 \pm .04$ | $80.9 \pm 0.6$ | $0.81 \pm .04$ | $83.2 \pm 1.1$ | $\mathbf{0.90 \pm .01}$ | $\mathbf{83.4 \pm 0.3}$ | |
| Citeseer | 1.3% | $0.38 \pm .11$ | $71.9 \pm 0.8$ | $0.15 \pm .07$ | $70.7 \pm 0.9$ | $0.68 \pm .03$ | $71.3 \pm 1.2$ | $\mathbf{0.79 \pm .01}$ | $\mathbf{73.1 \pm 0.2}$ | |
| | 2.6% | $0.56 \pm .06$ | $72.2 \pm 0.4$ | $0.29 \pm .05$ | $70.8 \pm 0.5$ | $0.79 \pm .05$ | $71.5 \pm 0.7$ | $\mathbf{0.83 \pm .02}$ | $\mathbf{73.3 \pm 0.5}$ | $73.7 \pm 0.6$ |
| | 5.2% | $0.71 \pm .05$ | $73.0 \pm 0.3$ | $0.35 \pm .08$ | $70.2 \pm 0.4$ | $0.83 \pm .03$ | $71.1 \pm 0.8$ | $\mathbf{0.89 \pm .02}$ | $\mathbf{73.4 \pm 0.4}$ | |
| Ogbn-arxiv | 0.1% | $0.59 \pm .08$ | $70.1 \pm 1.7$ | $0.39 \pm .06$ | $69.8 \pm 1.4$ | $0.59 \pm .07$ | $70.3 \pm 1.4$ | $\mathbf{0.77 \pm .04}$ | $\mathbf{71.9 \pm 0.8}$ | |
| | 0.25% | $0.63 \pm .05$ | $70.3 \pm 1.3$ | $0.44 \pm .03$ | $70.1 \pm 0.7$ | $0.64 \pm .05$ | $70.5 \pm 1.0$ | $\mathbf{0.83 \pm .03}$ | $\mathbf{72.4 \pm 1.0}$ | $73.2 \pm 0.8$ |
| | 0.5% | $0.68 \pm .07$ | $70.9 \pm 1.0$ | $0.47 \pm .05$ | $70.0 \pm 0.7$ | $0.67 \pm .05$ | $71.1 \pm 0.6$ | $\mathbf{0.88 \pm .03}$ | $\mathbf{72.6 \pm 0.6}$ | |
| Reddit | 0.1% | $0.42 \pm .09$ | $92.1 \pm 1.6$ | $0.39 \pm .04$ | $90.9 \pm 0.8$ | $0.53 \pm .06$ | $90.9 \pm 1.7$ | $\mathbf{0.79 \pm .03}$ | $\mathbf{92.1 \pm 0.9}$ | |
| | 0.25% | $0.50 \pm .06$ | $92.7 \pm 1.3$ | $0.41 \pm .05$ | $90.9 \pm 0.5$ | $0.61 \pm .04$ | $91.2 \pm 1.2$ | $\mathbf{0.83 \pm .01}$ | $\mathbf{92.9 \pm 0.7}$ | $94.1 \pm 0.7$ |
| | 0.5% | $0.58 \pm .06$ | $92.8 \pm 0.7$ | $0.42 \pm .03$ | $91.5 \pm 0.6$ | $0.66 \pm .02$ | $92.1 \pm 0.9$ | $\mathbf{0.87 \pm .01}$ | $\mathbf{93.1 \pm 0.5}$ | |

Table 3: The search time and test performance of the best architecture find by NAS methods on the condensed datasets. We consider two NAS algorithms: (1) the differentiable NAS algorithm DARTS-PT and (2) REINFORCE without parameter-sharing.

| NAS Algorithm | Random | | DC | | HCDC | | Original | |
|---|---|---|---|---|---|---|---|---|
| | Time (sec) | Perf. (%) | Time (sec) | Perf. (%) | Time (sec) | Perf. (%) | Time (sec) | Perf. (%) |
| DARTS-PT | 37.1 | $89.4 \pm 0.3$ | 39.2 | $85.2 \pm 1.9$ | | | | |
| REINFORCE | 166 | $88.1 \pm 1.8$ | 105 | $80.1 \pm 6.5$ | | | | |

**Preserving architecture performance ranking on images.** We follow the practice of (Cui et al., 2022) and construct the search space by sampling 100 networks from NAS-Bench-201 (Dong & Yang, 2020), which contains the ground-truth performance of 15,625 networks. All models are trained on CIFAR-10 or CIFAR-100 for 50 epochs under five random seeds and ranked according to their average accuracy on a held-out validation set of 10K images. As a common practice in NAS (Liu et al., 2018), we reduce the number of repeated blocks in all architecture from 15 to 3 during the search phase, as deep models are hard to train on the small condensed datasets. We consider three coreset baselines, including uniform random sampling, K-Center (Farahani & Hekmatfar, 2009), and Herding (Welling, 2009) coresets, as well as five standard condensation baselines, including dataset condensation (DC) (Zhao et al., 2020), differentiable siamese augmentation (DSA) (Zhao & Bilen, 2021a), distribution matching

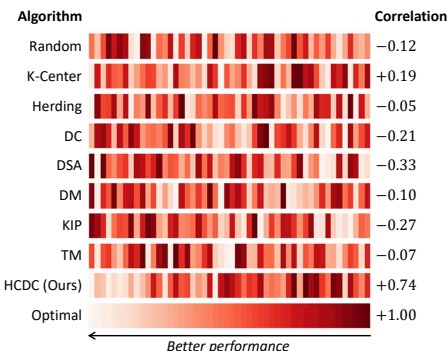

Figure 2: Visualization of the performance rankings of architectures (subsampled from the search space) evaluated on different condensed datasets. Colors indicate the performance ranking on the original dataset, while lighter shades refer to better performance. Spearman's rank correlations are shown on the right.

(DM) (Zhao & Bilen, 2021b), Kernel Inducing Point (KIP) (Nguyen et al., 2020; 2021), and Training Trajectory Matching (TM) (Cazenavette et al., 2022). For the coreset and condensation baselines, we randomly split the condensed dataset to obtain the condensed validation data while keeping the train-validation split ratio. We subsample or compress the original dataset to 50 images per class for all baselines. As shown in Table 1, our HCDC is much better at preserving the performance ranking of architectures compared to all other coreset and condensation methods. At the same time, HCDC also consistently attains better-selected architectures' performance, which implies the HCDC condensed datasets are reliable proxies of the original datasets for architecture search.

**Speeding up architecture search on images.** We then combine HCDC with some off-the-shelf NAS algorithms to demonstrate the efficiency gain when evaluated on the proxy condensed dataset. We consider two NAS algorithms: DARTS-PT (Wang et al., 2020), which is a parameter-sharing based differentiable NAS algorithm, and REINFORCE (Williams, 1992), which is a reinforcement learning algorithm without parameter sharing. In Table 3, we see all coreset/condensation baselines can bring significant speed-ups to the NAS algorithms since the models are trained on the small proxy datasets. Same as under the grid search setup, the test performance of the selected architecture on the HCDC condensed dataset is consistently higher. Here a small search space of 100 sampled architectures is used as in Table 3 and we expect even higher efficiency gain on larger search spaces.

In Fig. 2, we directly visualize the performance rankings of architectures on different condensed datasets. Each color slice indicates one architecture and and are re-ordered with the ranking from the condensation algorithm. rows that are more similar to the 'optimal' gradient indicate that the algorithm is ranking the architectures similarly to the optimal ranking. The best architectures with the HCDC algorithm are among the best in the original dataset.

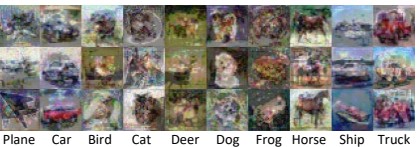

Plane  Car  Bird  Cat  Deer  Dog  Frog  Horse  Ship  Truck

Figure 3: Visualization of some example condensed validation set images using our HCDC algorithm on CIFAR-10.

**Finding the best convolution filter on graphs.** We now consider the search space of graph neural networks' (GNN) convolution filters, which is intrinsically continuous (i.e., defined by a few continuous hyperparameters which parameterize the convolution filter, see Section 5 for details). Our goal is to speed up the selection of the best-suited convolution filter design on large graphs. We consider 2-layer message-passing GNNs whose convolution matrix is a truncated sum of powers of the graph Laplacian; see Appendix J. Four node classification graph benchmarks are used, including two small graphs (Cora and Citeseer) and two large graphs (Ogbn-arxiv and Reddit) with more than 100K nodes. To compute the Spearman's rank correlations, we sample 80 hyperparameter setups from the search space and compare their performance rankings. We test HCDC against three baselines: (1) Random: random uniform sampling of nodes and find their induced subgraph, (2) GCond-X: graph condensation (Jin et al., 2021) but fix the synthetic adjacency to the identity matrix, (3) GCond: graph condensation which also learns the adjacency. The whole graph performance is oracle and shows the best possible test performance when the convolution filter is optimized on the original datasets using hypergradient-based method (Lorraine et al., 2020). In Table 2, we see HCDC consistently outperforms the other approaches, and the test performance of selected architecture is close to the ground-truth optimal.

**Speeding up off-the-shelf graph architecture search algorithms.** Finally, we demonstrate HCDC can speed up off-the-shelf graph architecture search methods. We use graph NAS (Gao et al., 2019) on Ogbn-arxiv with a compression ratio of $c_{\text{train}}/n = 0.5\%$, where $n$ is the size of the original graph, and $c_{\text{train}}$ is the number of training nodes in the condensed graph. The search space of GNN architectures is the same as in (Gao et al., 2019), where various attention and aggregation functions are incorporated. We plot the best test performance of searched architecture versus the search time in Fig. 4. We see that when evaluated on the dataset condensed by HCDC, the search algorithm finds the better architectures

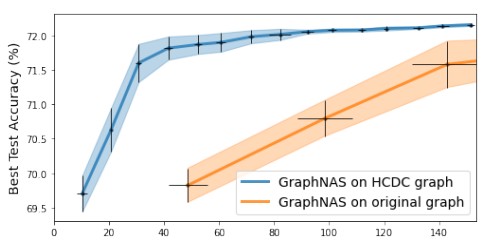

Figure 4: Speed-up of graph NAS's search process, when evaluated on the small proxy dataset condensed by HCDC.

much faster. This efficiency gain provided by the small proxy dataset is orthogonal to the design of search strategies and should be applied to any type of data, including graph and images.

## 8  CONCLUSION

We propose a hyperparameter calibration formulation for dataset condensation to preserve the outcome of hyperparameter optimization, which is then solved by aligning the hyperparameter gradients. We demonstrate both theoretically and experimentally that HCDC can effectively preserve the validation performance rankings of architectures and accelerate the hyperparameter/architecture search on images and graphs. The overall performance of HCDC can be affected by (1) how the differentiable NAS model used for condensation generalizes to unseen architectures, (2) where we align hypergradients in the search space, (3) how we learn the synthetic training set, (4) how we parameterize the synthetic dataset, and we leave the exploration of these design choices to future work. We hope our work opens up a promising avenue for speeding up hyperparameter/architecture search by dataset compression.

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
