# A    A DETAILED DIAGRAM OF THE PROPOSED HCDC

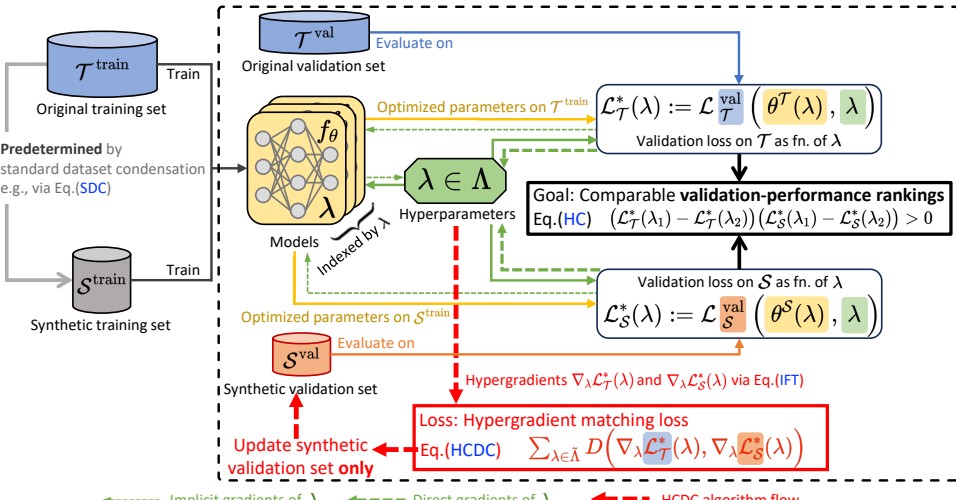

Figure 5: Hyperparameter Calibrated Dataset Condensation (HCDC) aims to find a small validation dataset such that the validation-performance rankings of the models with different hyperparameters are comparable to the large original dataset's $\big($Eq. (HC) in the "Goal" box$\big)$. Our method realizes this goal by learning the synthetic validation set to match the hypergradients w.r.t the hyperparameters $\big($Eq. (HCDC) in the "Loss" box$\big)$. **Our contribution is depicted within the big black dashed box.** The algorithm flow is indicated through the red dashed arrows. Solid arrows (blue, yellow and green) indicate forward passes. To calculate the hypergradients with respect to hyperparameters $\lambda$, we backpropagate to compute both implicit/direct gradients (thin/thick green dashed arrows). The synthetic training set is predetermined by any standard dataset condensation (SDC) methods (e.g., Eq. (SDC)). The synthetic training and validation datasets obtained can later be used for hyperparameter search using only a fraction of the original computational load.

# B    MORE RELATED WORK

This section contains extensive discussions of many related areas, which cannot be fitted into the main paper due to the page limit.

## B.1    DATASET CONDENSATION AND CORESET SELECTION

Firstly, we review the two main approaches to reducing the training set size while preserving model performance.

**Dataset condensation** (or distillation) is first proposed in (Wang et al., 2018) as a learning-to-learn problem by formulating the network parameters as a function of synthetic data and learning them through the network parameters to minimize the training loss over the original data. However, the nested-loop optimization precludes it from scaling up to large-scale in-the-wild datasets. Zhao et al. (2020) alleviate this issue by enforcing the gradients of the synthetic samples w.r.t. the network weights to approach those of the original data, which successfully alleviates the expensive unrolling of the computational graph. Based on the meta-learning formulation in (Wang et al., 2018), Bohdal et al. (2020) and Nguyen et al. (2020; 2021) propose to simplify the inner-loop optimization of a classification model by training with ridge regression which has a closed-form solution, while Such et al. (2020) model the synthetic data using a generative network. To improve the data efficiency of synthetic samples in gradient-matching algorithm, Zhao & Bilen (2021a) apply differentiable Siamese augmentation, and Kim et al. (2022) introduce efficient synthetic-data parametrization. Recently, a new distribution-matching framework (Zhao & Bilen, 2021b) proposes to match the hidden features rather than the gradients for fast optimization but may suffer from performance degradation compared to gradient-matching (Zhao & Bilen, 2021b), where Kim et al. (2022) provide some interpretation.

**Coreset selection** methods choose samples that are important for training based on heuristic criteria, for example, minimizing the distance between coreset and whole-dataset centers (Chen et al., 2010; Rebuffi et al., 2017), maximizing the diversity of selected samples in the gradient space (Aljundi et al., 2019), discovering cluster centers (Sener & Savarese, 2018), and choosing samples with the largest negative implicit gradient (Borsos et al., 2020). *Forgetting* (Toneva et al., 2018) measures the forgetfulness of trained samples and drops those that are not easy to forget. *GraNd* (Paul et al., 2021) selects the training samples that contribute most to the training loss in the first few epochs. *Prism* (Kothawade et al., 2022) select samples to maximize submodular set-functions, which are combinatorial generalizations of entropy measures (Iyer et al., 2021). Recent benchmark (Guo et al., 2022) of a variety of coreset selection methods for image classification indicates *Forgetting*, *GraNd*, and *Prism* are among the best-performing corset methods but still evidently underperform the dataset condensation baselines. Although coreset selection can be very efficient, most of the methods above suffer from three major limitations: (1) their performance is upper-bounded by the information in the selected samples; (2) most of them do not directly optimize the synthetic samples to preserve the model performance; and (3) most of them select samples incrementally and greedily, which is short-sighted.

## B.2 Implicit Differentiation and Differentiable NAS

Secondly, we list two relevant areas to this work: implicit differentiation methods based on the implicit function theorem (IFT), and the differentiable neural architecture search (NAS) algorithms.

**Implicit differentiation** methods apply the implicit function theorem (IFT) to the nested-optimization problems (Ochs et al., 2015; Wang et al., 2019). The IFT requires inverting the training Hessian with respect to the network weights, where early work either computes the inverse explicitly (Bengio, 2000; Larsen et al., 1996) or approximates it as the identity matrix (Luketina et al., 2016). Conjugate gradient (CG) is applied to invert the Hessian approximately (Pedregosa, 2016), but is difficult to scale to deep networks. Several methods have been proposed to efficiently approximate Hessian inverse, for example, 1-step unrolled differentiation (Luketina et al., 2016), Fisher information matrix (Larsen et al., 1996), NN-structure aided Kronecker-factored inversion (Martens & Grosse, 2015). Lorraine et al. (2020) use the Neumann inverse approximation, which is a stable alternative to CG (Shaban et al., 2019) and successfully scale gradient-based bilevel-optimization to large networks with constant memory constraint. It is shown that unrolling differentiation around locally optimal weights for $i$ steps is equivalent to approximating the Neumann series inverse approximation up to the first $i$ terms.

**Differentiable NAS** methods, e.g., DARTS (Liu et al., 2018) explores the possibility of transforming the discrete neural architecture space into a continuously differentiable form and further uses gradient optimization to search the neural architecture. DARTS follows a cell-based search space (Zoph et al., 2018) and continuously relaxes the original discrete search strategy. Despite its simplicity, several works cast double on the effectiveness of DARTS (Li & Talwalkar, 2020; Zela et al., 2019). SNAS (Xie et al., 2018) points out that DARTS suffers from the unbounded bias issue towards its objective, and it remodels the NAS and leverages the Gumbel-softmax trick (Jang et al., 2017; Maddison et al., 2017) to learn the exact architecture parameter. Differentiable NAS techniques have also been applied to graphs to design data-specific GNN architectures (Wang et al., 2021; Huan et al., 2021) automatically.

## B.3 Graph Reduction

Lastly, when we apply HCDC to graph data, it relates to the graph reduction method for graph neural network training, which we summarize as follows.

**Graph coreset selection** is a non-trivial generalization of the above method coreset methods given the non-*iid* nature of graph nodes and the non-linearity nature of GNNs. The very few off-the-shelf graph coreset algorithms are designed for graph clustering (Baker et al., 2020; Braverman et al., 2021) and are not optimal for the training of GNNs.

**Graph condensation** (Jin et al., 2021) achieves the state-of-the-art performance for preserving GNNs' performance on the simplified graph. However, Jin et al. (2021) only adapt the gradient-matching algorithm of dataset condensation Zhao et al. (2020) to graph data, together with an MLP-based

generative model for edges (Anand & Huang, 2018; Simonovsky & Komodakis, 2018), leaving out several major issues on efficiency, performance, and generalizability. Subsequent work aims to apply the more efficient distribution-matching algorithm (Zhao & Bilen, 2021b; Wang et al., 2022) of dataset condensation to graph (Liu et al., 2022) or speed up gradient-matching graph condensation by reducing the number of gradient-matching-steps (Jin et al., 2022). While the efficiency issue of graph condensation is mitigated (Jin et al., 2022), the performance degradation on medium- and large-sized graphs still renders graph condensation practically meaningless. Our HCDC is specifically designed for repeated training in architecture search, which is, in contrast, well-motivated.

**Graph sampling** methods (Chiang et al., 2019; Zeng et al., 2019) can be as simple as uniformly sampling a set of nodes and finding their induced subgraph, which is understood as a graph-counterpart of uniform sampling of *iid* samples. However, most of the present graph sampling algorithms (e.g., ClusterGCN (Chiang et al., 2019) and GraphSAINT (Zeng et al., 2019)) are designed for sampling multiple subgraphs (mini-batches), which form a cover of the original graph for training GNNs with memory constraint. Therefore those graph mini-batch sampling algorithms are effectively graph partitioning algorithms and not optimized to find just one representative subgraph.

**Graph sparsification** (Batson et al., 2013; Satuluri et al., 2011) and **graph coarsening** (Loukas & Vandergheynst, 2018; Loukas, 2019; Huang et al., 2021; Cai et al., 2020) algorithms are usually designed to preserve specific graph properties like graph spectrum and graph clustering. Such objectives are often not aligned with the optimization of downstream GNNs and are shown to be sub-optimal in preserving the information to train GNNs well (Jin et al., 2021).

## C PRELIMINARIES ON GRAPH HYPERPARAMETER SEARCH

HCDC applies to not only image data but also graph data and graph neural networks (GNN). In Section 7, we use HCDC to search for the best convolution filter in message-passing GNNs. In this section, we give the backgrounds of the graph-related experiments.

### C.1 NODE CLASSIFICATION AND GNNS

Node classification on a graph considers a graph $\mathcal{T} = (A, X, \mathbf{y})$ with adjacency matrix $A \in \{0,1\}^{n \times n}$, node features $X \in \mathbb{R}^{n \times d}$, node class labels $\mathbf{y}$, and mutually disjoint node-splits $V_{\text{train}} \bigcup V_{\text{val}} \bigcup V_{\text{test}} = [n]$. Using a *graph neural network* (GNN) $f_{\theta,\lambda} : \mathbb{R}_{\geq 0}^{n \times n} \times \mathbb{R}^{n \times d} \to \mathbb{R}^{n \times K}$, where $\theta \in \Theta$ denotes the parameters and $\lambda \in \Lambda$ denotes the hyper-parameters (if they exist), we aim to find $\theta^{\mathcal{T}} = \arg \min_\theta \mathcal{L}_{\mathcal{T}}^{\text{train}}(\theta, \lambda)$, where $\mathcal{L}_{\mathcal{T}}^{\text{train}}(\theta, \lambda) := \sum_{i \in V_{\text{train}}} \ell([f_{\theta,\lambda}(A, X)]_i, y_i)$ and $\ell(\hat{y}, y)$ is the cross-entropy loss. The node classification loss $\mathcal{L}_{\mathcal{T}}^{\text{train}}(\theta, \lambda)$ is under the *transductive* setting, which can be easily generalized to the *inductive* setting by assuming only $\{A_{ij} \mid i, j \in V_{\text{train}}\}$ and $\{X_i \mid i \in V_{\text{train}}\}$ are used during training.

### C.2 ADDITIONAL DOWNSTREAM TASKS

We have defined the downstream task this paper mainly focuses on, node classification on graphs. Where we are given a graph $\mathcal{T} = (A, X, \mathbf{y})$ with adjacency matrix $A \in \{0,1\}^{n \times n}$, node features $X \in \mathbb{R}^{n \times d}$, node class labels $\mathbf{y} \in [K]^n$, and mutually disjoint node-splits $V_{\text{train}} \bigcup V_{\text{val}} \bigcup V_{\text{test}} = [n]$, and the goal is to predict the node labels.

Here, we show that the settings above can also be used to describe per-pixel classification on images (e.g., for semantic segmentation) where CNNs are usually used. For *per-pixel classification*, we are given a set of $\mathfrak{n}$ images of size $w \times h$, so the pixel values of the $j$-th image can be formatted as a tensor $\mathfrak{X}_j \in \mathbb{R}^{w \times h \times c}$ if there are $c$ channels. We are also given the pixel labels $\mathfrak{Y}_j \in [K]^{w \times h}$ for each image $j \in [\mathfrak{n}]$ and the mutually disjoint image-splits $I_{train} \bigcup I_{val} \bigcup I_{test} = [\mathfrak{n}]$. Clearly, we can reshape the pixel values and pixel labels of the $j$-th image to $wh \times c$ and $wh$, respectively, and concatenate those matrices from all images. Following this, denoting $n = \mathfrak{n}wh$, we obtain the concatenated pixel value matrix $X \in \mathbb{R}^{n \times c}$ and the concatenated pixel label vector $\mathbf{y} \in [K]^n$. The image-splits are translated into pixel-level splits where $V_{train} = \{i \mid (j-1)wh \leq i \leq jwh, j \in I_{trian}\}$ (similar for $V_{val}$ and $V_{test}$) and $V_{train} \bigcup V_{val} \bigcup V_{test} = [n]$. We can also define the auxiliary adjacency matrix $A \in \{0,1\}^{n \times n}$ on the $n = \mathfrak{n}wh$ pixels, where $A$ is block diagonal $A = \text{diag}(A_1, \ldots, A_\mathfrak{n})$ and $A_j \in \{0,1\}^{wh \times wh}$ is the assumed adjacency (e.g., a two-dimensional grid) of the $j$-th image.

### C.3 GRAPH NEURAL NETWORK MODELS

Here we mainly focus on graph neural networks (GNNs) $f_{\theta,\lambda} : \mathbb{R}_{\geq 0}^{n \times n} \times \mathbb{R}^{n \times d} \to \mathbb{R}^{n \times K}$, where $\theta \in \Theta$ denotes the parameters, and $\lambda \in \Lambda$ denotes the hyperparameters. In Section 2, we have seen that most GNNs can be interpreted as iterative convolution/message passing over nodes (Ding et al., 2021; Balcilar et al., 2021) where $X^{(0)} = X$ and $f(A, X) = X^{(L)}$, and for $l \in [L]$, the update-rule is,

$$X^{(l+1)} = \sigma\Big(C_{\alpha^{(l)}}(A)X^{(l)}W^{(l)}\Big), \tag{4}$$

where $C_{\alpha^{(l)}}(A)$ is the convolution matrix parametrized by $\alpha^{(l)}$, $W^{(l)}$ is the learnable linear weights, and $\sigma(\cdot)$ denotes the non-linearity. Thus the parameters $\theta$ consists of all $\alpha$'s (if they exist) and $W$'s, i.e., $\theta = [\alpha^{(0)}, \dots, \alpha^{(L-1)}, W^{(0)}, \dots, W^{(L-1)}]$.

More specifically, it is possible for GNNs to have more than one convolution filter per layer (Ding et al., 2021; Balcilar et al., 2021) and we may generalize Eq. (4) to,

$$X^{(l+1)} = \sigma\left(\sum_{i=1}^{p} C_{\alpha^{(l,i)}}^{(i)}(A)X^{(l)}W^{(l,i)}\right). \tag{5}$$

Within this common framework, GNNs differ from each other by choice of convolution filters $\{C^{(i)}\}$, which can be either fixed or learnable. If $C^{(i)}$ is fixed, there is no parameters $\alpha^{(l,i)}$ for any $l \in [L]$. If $C^{(i)}$ is learnable, the convolution matrix relies on the learnable parameters $\alpha^{(l,i)}$ and can be different in each layer (thus should be denoted as $C^{(l,i)}$). Usually, for GNNs, the convolution matrix depends on the parameters in two possible ways: (1) the convolution matrix $C^{(l,i)}$ is scaled by the scalar parameter $\alpha^{(l,i)} \in \mathbb{R}$, i.e., $C^{(l,i)} = \alpha^{(l,i)}\mathfrak{C}^{(i)}$ (e.g., GIN (Xu et al., 2018), ChebNet (Defferrard et al., 2016), and SIGN (Frasca et al., 2020)); or (2) the convolution matrix is constructed by node-level self-attentions $[C^{(l,i)}]_{ij} = h_{\alpha^{(l,i)}}\big(X_{i,:}^{(l)}, X_{j,:}^{(l)}\big)[\mathfrak{C}^{(i)}]_{i,j}$ (e.g., GAT (Veličković et al., 2018), Graph Transformers (Rong et al., 2020; Puny et al., 2020; Zhang et al., 2020)). Based on (Ding et al., 2021; Balcilar et al., 2021), we summarize the popular GNNs reformulated into the convolution over nodes / message-passing formula (Eq. (5)) in Table 4.

Convolutional neural networks can also be reformulated into the form of Eq. (5). For simplicity, we only consider a one-dimensional convolution neural network (1D-CNN), and the generalization to 2D/3D-CNNs is trivial. If we denote the constant cyclic permutation matrix (which corresponds to a unit shift) as $P \in \mathbb{R}^{n \times n}$, the update rule of a 1D-CNN with kernel size $(2K + 1), K \geq 0$ can be written as,

$$X^{(l+1)} = \sigma\Big(\sum_{k=-K}^{k=K} \alpha_k P^k X^{(l)}W^{(l,k)}\Big). \tag{6}$$

We will use this common convolution formula of GNNs (Eq. (5)) and 1D-CNNs (Eq. (6)) in Appendix C.5 and Proposition 2.

### C.4 HCDC IS APPLICABLE TO VARIOUS DATA, TASKS, AND MODELS

In Appendices C.2 and C.3, we have discussed the formal definition of two possible tasks (1) node classification on graphs and (2) per-pixel classification on images, and reformulated many popular GNNs and CNNs into a general convolution form (Eqs. (5) and (6)). However, we want to note that the application of dataset condensation methods (including the standard dataset condensation (Wang et al., 2018; Zhao et al., 2020; Zhao & Bilen, 2021b) and our HCDC) is not limited by the specific types of data, tasks, and models.

For HCDC, we can follow the conventions in (Zhao et al., 2020) to define the train/validation losses on *iid* samples and define the notion of dataset condensation as learning a smaller synthetic dataset with less number of samples. Here we leave the readers to (Zhao et al., 2020) for formal definitions of condensation on datasets with *iid* samples.

More generally speaking, our HCDC can be applied as long as (1) the train and validation losses, i.e., $\mathcal{L}_{\mathcal{T}}^{train}(\theta, \lambda)$ and $\mathcal{L}_{\mathcal{T}}^{val}(\theta, \lambda)$ can be defined (as functions of the parameters and hyperparameters); and (2) we have a well-defined notion of the learnable synthetic dataset $\mathcal{S}$, (e.g., which includes

Table 4: Summary of GNNs formulated as generalized graph convolution.

| Model Name | Design Idea | Conv. Matrix Type | # of Conv. | Convolution Matrix |
|---|---|---|---|---|
| GCN[1] (Kipf & Welling, 2016) | Spatial Conv. | Fixed | 1 | $C = \widetilde{D}^{-1/2}\widetilde{A}\widetilde{D}^{-1/2}$ |
| SAGE-Mean[2] (Hamilton et al., 2017) | Message Passing | Fixed | 2 | $\begin{cases} C^{(1)} = I_n \\ C^{(2)} = D^{-1}A \end{cases}$ |
| GAT[3] (Veličković et al., 2018) | Self-Attention | Learnable | # of heads | $\begin{cases} \mathfrak{C}^{(s)} = A + I_n \text{ and} \\ h_{\boldsymbol{a}^{(l,s)}}^{(s)}(X_{i,:}^{(l)}, X_{j,:}^{(l)}) = \exp\big(\text{LeakyReLU}( \\ \quad (X_{i,:}^{(l)}W^{(l,s)} \parallel X_{j,:}^{(l)}W^{(l,s)}) \cdot \boldsymbol{a}^{(l,s)})\big) \end{cases}$ |
| GIN[1] (Xu et al., 2018) | WL-Test | Fixed + Learnable | 2 | $\begin{cases} C^{(1)} = A \\ \mathfrak{C}^{(2)} = I_n \text{ and } h_{\epsilon^{(l)}}^{(2)} = 1 + \epsilon^{(l)} \end{cases}$ |
| SGC[2] (Defferrard et al., 2016) | Spectral Conv. | Learnable | order of poly. | $\begin{cases} \mathfrak{C}^{(1)} = I_n, \mathfrak{C}^{(2)} = 2L/\lambda_{\max} - I_n, \\ \mathfrak{C}^{(s)} = 2\mathfrak{C}^{(2)}\mathfrak{C}^{(s-1)} - \mathfrak{C}^{(s-2)} \\ \text{and } h_{\theta^{(s)}}^{(s)} = \theta^{(s)} \end{cases}$ |
| ChebNet[2] (Defferrard et al., 2016) | Spectral Conv. | Learnable | order of poly. | $\begin{cases} \mathfrak{C}^{(1)} = I_n, \mathfrak{C}^{(2)} = 2L/\lambda_{\max} - I_n, \\ \mathfrak{C}^{(s)} = 2\mathfrak{C}^{(2)}\mathfrak{C}^{(s-1)} - \mathfrak{C}^{(s-2)} \\ \text{and } h_{\theta^{(s)}}^{(s)} = \theta^{(s)} \end{cases}$ |
| GDC[3] (Klicpera et al., 2019) | Diffusion | Fixed | 1 | $C = S$ |
| Graph Transformers[4] (Rong et al., 2020) | Self-Attention | Learnable | # of heads | $\begin{cases} \mathfrak{C}_{i,j}^{(s)} = 1 \text{ and } h_{(W_Q^{(l,s)}, W_K^{(l,s)})}^{(s)}(X_{i,:}^{(l)}, X_{j,:}^{(l)}) \\ = \exp\big(\frac{1}{\sqrt{d_{k,l}}}(X_{i,:}^{(l)}W_Q^{(l,s)})(X_{j,:}^{(l)}W_K^{(l,s)})^{\mathsf{T}}\big) \end{cases}$ |

[1] Where $\widetilde{A} = A + I_n$, $\widetilde{D} = D + I_n$. [2] $C^{(2)}$ represents mean aggregator. Weight matrix in (Hamilton et al., 2017) is $W^{(l)} = W^{(l,1)} \parallel W^{(l,2)}$. [3] Need row-wise normalization. $C_{i,j}^{(l,s)}$ is non-zero if and only if $A_{i,j} = 1$, thus GAT follows direct-neighbor aggregation. [4] The weight matrices of the two convolution supports are the same, $W^{(l,1)} = W^{(l,2)}$. [5] Where normalized Laplacian $L = I_n - D^{-1/2}AD^{-1/2}$ and $\lambda_{\max}$ is its largest eigenvalue, which can be approximated as 2 for a large graph. [6] Where $S$ is the diffusion matrix $S = \sum_{k=0}^{\infty} \theta_k \boldsymbol{T}^k$, for example, decaying weights $\theta_k = e^{-t}\frac{t^k}{k!}$ and transition matrix $\boldsymbol{T} = \widetilde{D}^{-1/2}\widetilde{A}\widetilde{D}^{-1/2}$. [7] Need row-wise normalization. Only describes the global self-attention layer, where $W_Q^{(l,s)}, W_Q^{(l,s)} \in \mathbb{R}^{f_l, d_{k,l}}$ are weight matrices which compute the queries and keys vectors. In contrast to GAT, all entries of $\mathfrak{C}_{i,j}^{(l,s)}$ are non-zero. Different design of Graph Transformers (Puny et al., 2020; Rong et al., 2020; Zhang et al., 2020) use graph adjacency information in different ways and is not characterized here; see the original papers for details.

prior-knowledge like what is the format of the synthetic data in $\mathcal{S}$ and how the same model $f_{\theta,\lambda}$ is applied).

## C.5 THE LINEAR CONVOLUTION REGRESSION PROBLEM ON GRAPH

For the ease of theoretical analysis, in Lemma 5 and Propositions 2 to 4 we consider a simplified *linear convolution regression problem* as follows,

$$\theta^{\mathcal{T}} = \arg\min_{\theta=[\alpha,W]} \|C_\alpha(A)\,XW - \mathbf{y}\|^2 \tag{7}$$

where we are given continuous labels $\mathbf{y}$ and use sum-of-squares loss $\ell(\hat{y}, y) = \|\hat{y} - y\|_2^2$ instead of the cross entropy loss used for node/pixel classification. We also assume a linear GNN/CNN $f_{\theta=[\alpha,W]}(A, X) = C_\alpha(A)XW$ is used, where $C_\alpha(A)$ is the *convolution matrix* which depends on the adjacency matrix $A$ and the parameters $\alpha \in \mathbb{R}^p$, and $W$ is the learnable linear weights with $d$ elements (hence, the complete parameters consist of two parts, i.e., $\theta = [\alpha, W]$).

As explained in Appendix C.3, this *linear convolution* model $f_{\theta=[\alpha,W]}(A, X) = C_\alpha(A)XW$ already generalizes a wide variety of GNNs and CNNs. For example, it can represent the (single-layer) graph convolution network (GCN) (Kipf & Welling, 2016) whose convolution matrix is defined as $C(A) = \tilde{D}^{-\frac{1}{2}}\tilde{A}\tilde{D}^{-\frac{1}{2}}$ where $\tilde{A}$ and $\tilde{D}$ are the "self-loop-added" adjacency and degree matrix (for GCNN there is no learnable parameters in $C_(A)$ so we omit $\alpha$). It also generalizes the one-dimensional *convolution neural network* (1D-CNN), where the convolution matrix is $C_\alpha(A) = \sum_{k=-K}^{k=K}[\theta]_k P^k$ and $P$ is the cyclic permutation matrix correspond to a unit shift.

It is important to note that although we considered this simplified *linear convolution regression problem* in some of our theoretical results, which is both convex and linear. We argue that most of the theoretical phenomena reflected by Lemma 5 and Propositions 2 to 4 can be generalized to the general non-convex losses and non-linear models; see Appendix F.4 for the corresponding discussions.

# D    STANDARD DATASET CONDENSATION IS PROBLEMATIC ACROSS GNNS

In this section, we analyze that standard dataset condensation (SDC) is especially problematic when applied to graphs and GNNs, due to the poor generalizability across GNNs with different convolution filters.

For ease of theoretical discussions, in this subsection, we consider single-layer message-passing GNNs. Message passing GNNs can be interpreted as iterative convolution over nodes (i.e., *message passing*) (Ding et al., 2021) where $X^{(0)} = X$, $X^{(l+1)} = \sigma(C_{\alpha^{(l)}}(A)X^{(l)}W^{(l)})$ for $l \in [L]$, and $f(A, X) = X^{(L)}$, where $C_{\alpha^{(l)}}(A)$ is the convolution matrix parametrized by $\alpha^{(l)}$, $W^{(l)}$ is the learnable linear weights, and $\sigma(\cdot)$ denotes the non-linearity. One-dimensional convolution neural networks (1D-CNNs) can be expressed by a similar formula, $f(X) = (\sum_{k=-K}^{k=K} \alpha^{(k)} P^k)XW$, parameterized by $\theta = [\alpha, W]$ where $\alpha = [\alpha^{(-K)}, \dots, \alpha^{(K)}]$. $P$ is the cyclic permutation matrix (of a unit shift). The kernel size is $(2K + 1), K \geq 0$; see Appendix C.3 for details.

Despite the success of the gradient matching algorithm in preserving the model performance when trained on the condensed dataset (Wang et al., 2022), it naturally overfits the model $f_{\theta,\lambda}$ used during condensation and generalizes poorly to others. There is no guarantee that the condensed synthetic data $\mathcal{S}^*$ which minimizes the objective (Eq. (2)) for a specific model $f_{\theta,\lambda}$ (marked by its hyperparameter $\lambda$) can generalize well to other models $f_{\theta,\lambda'}$ where $\lambda' \neq \lambda$. We aim to demonstrate that this overfitting issue can be much more *severe on graphs* than on images, where our main theoretical results can be informally summarized as follows.

**Informal Proposition.** *Standard dataset condensation using gradient matching algorithm (Eq. (2)) is problematic across GNNs. The condensed graph using a single-layer message passing GNN may fail to generalize to the other GNNs with a different convolution matrix.*

We first show the successful generalization of SDC across one-dimensional *convolution neural networks* (1D-CNN). Then, we show a contrary result on GNNs: failed generalization of SDC across GNNs. These theoretical analyses demonstrate the hardness of data condensation on graphs. Our analysis is based on the achievability condition of a gradient matching objective; see Assumption 1 in Appendix D.

In Lemma 5 of Appendix F.1, under least square regression with linear GNN/CNN (see Appendix C.5 for formal definitions), if the standard dataset condensation GM objective is achievable, then the optimizer on the condensed dataset $\mathcal{S}$ is also optimal on the original dataset $\mathcal{T}$. Now, we study the generalizability of the condensed dataset across different models. We first show a successful generalization of SDC across different 1D-CNN networks; see Proposition 2 in Appendix D. As long as we use a 1D-CNN with a sufficiently large kernel size $K$ during condensation, we can generalize the condensed dataset to a wide range of models, i.e., 1D-CNNs with a kernel size $K' \leq K$.

However, we obtain a contrary result for GNNs regarding the generalizability of condensed datasets across models. Two dominant effects, which cause the failure of the condensed graph's ability to generalize across GNNs, are discovered.

Firstly, the learned adjacency $A'$ of the synthetic graph $\mathcal{S}$ can easily *overfit* the condensation objective (see Proposition 3), and thus can fail to maintain the characteristics of the original structure and distinguish between different architectures; see Proposition 3 in Appendix D for the theoretical result and Table 5 for relevant experiments.

Secondly, GNNs differ from each other mostly on the design of convolution filter $C(A)$, i.e., how the convolution weights $C$ depend on the adjacency information $A$. The convolution filter $C(A)$ used during condensation is a single biased point in "the space of convolutions"; see Fig. 6 for a visualization, thus there is a *mismatch* of inductive bias when transferring to a different GNN. These two effects lead to the obstacle when transferring the condensed graph across GNNs, which is formally characterized by Proposition 4 in Appendix D.

Proposition 4 provides an effective lower bound on the relative estimation error of optimal model parameters when a different convolution filter $C'(\cdot) \neq C(\cdot)$ is used.[1] According to the spectral characterization of convolution filters of GNNs (Table 1 of (Balcilar et al., 2021)), we can approximately

---

[1]If $C'(\cdot) = C(\cdot)$ Lemma 5 guarantees $W_{C'}^{\mathcal{S}} = W_{C'}^{\mathcal{T}}$ and the lower bound in Proposition 4 is 0.

| Ratio ($c/n$) | $A'$ learned | $A' = I_c$ |
|---|---|---|
| 0.05% | $59.2 \pm 1.1$ | $\mathbf{61.3} \pm 0.5$ |
| 0.25% | $63.2 \pm 0.3$ | $\mathbf{64.2} \pm 0.4$ |

(a) Test accuracy of graph condensation with learned or identity adjacency.

| Condense\Test | GCN | SGC ($K = 2$) | GIN |
|---|---|---|---|
| GCN | $\mathbf{60.3} \pm 0.3$ | $59.2 \pm 0.7$ | $42.2 \pm 4.3$ |
| SGC | $59.2 \pm 1.1$ | $\mathbf{60.5} \pm 0.6$ | $39.0 \pm 7.1$ |
| GIN | $47.5 \pm 3.6$ | $43.6 \pm 5.8$ | $\mathbf{59.1} \pm 1.1$ |

(b) Generalization accuracy of graphs condensed with different GNNs (row) across GNNs (column) under $c/n = 0.25\%$.

Table 5: Test accuracy of GNNs trained on condensed Ogbn-arxiv (Hu et al., 2020) graph verifying the two effects (Propositions 3 and 4) that hinders the generalization of the condensed graph across GNNs. (a) Condensed adjacency is overfitted to the SDC Objective, (b) Convolution filters and inductive bias mismatch across GNNs.

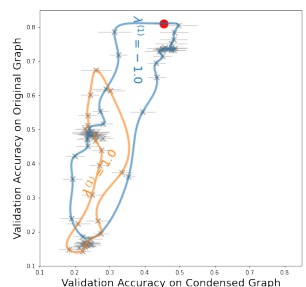

(a) Condense ratio $c/n = 0.2$

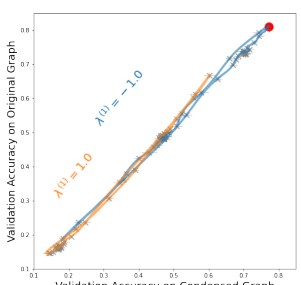

(b) Condense ratio $c/n = 0.8$

Figure 6: The manifold of GNNs with convolution filters $C_\lambda = I + \lambda^{(1)} L + \lambda^{(2)} \left( \frac{2}{\lambda_{\max}} L - I \right)$ (linear combination of first two orders of ChebNet (Defferrard et al., 2016), $\lambda$'s are hyperparameters; see Appendices C.3 and J) projected to the plane of validation accuracy on condensed (x-axis) and original (y-axis) graphs under two ratios $c/n$ on Cora (Yang et al., 2016). The GNN with $C = \left( \frac{2}{\lambda_{\max}} - 1 \right) L \propto L$ (red dot) is a biased point in this model space.

compute the maximum eigenvalue of $Q$ for some GNNs. For example, if we condense with $f^C$ graph isomorphism network (GIN-0) (Xu et al., 2018) but train $f^{C'}$ GCN on the condensed graph, we have $\|W_{C'}^{\mathcal{S}} - W_{C'}^{\mathcal{T}}\| / \|W_{C'}^{\mathcal{T}}\| \gtrsim \overline{\deg} + 1$ where $\overline{\deg}$ is the average node degree of the original graph. This large lower bound hints at the catastrophic failure when transferring across GIN and GCN; see Table 5.

**Assumption 1** (Achievability of a gradient matching Objective). *A gradient matching objective is defined to be achievable if there exists a non-degenerate trajectory $(\theta_t^{\mathcal{S}})_{t=0}^{T-1}$ (i.e., a trajectory that spans the entire parameter space $\Theta$, i.e., $\text{span}(\theta_0^{\mathcal{S}}, \dots, \theta_{T-1}^{\mathcal{S}}) \supseteq \Theta$), such that the gradient matching loss (the objective of Eq. (2) without expectation) on this trajectory is $0$.*

**Proposition 2** (Successful Generalization of SDC across 1D-CNNs). *Consider least-squares regression with one-dimensional linear convolution $f^{2K+1}(X)_\theta = \left( \sum_{k=-K}^{k=K} \alpha^{(k)} P^k \right) XW$ parameterized by $\theta = [\alpha, W]$ where $\alpha = [\alpha^{(-K)}, \dots, \alpha^{(K)}]$. $P$ is the cyclic permutation matrix (of a unit shift). The kernel size is $(2K + 1), K \geq 0$. If the gradient matching objective of $f^{2K+1}$ is achievable, then the condensed dataset $\mathcal{S}^*$ achieves the gradient matching objective on any trajectory $\{\theta_t'^{\mathcal{S}}\}_{t=0}^{T-1}$ for any linear convolution $f_{\theta'}^{2K'+1}$ with kernel size $(2K' + 1), K \geq K' \geq 0$.*

The intuition behind Proposition 2 is that the 1D-CNN of kernel size $(2K + 1)$ is a "supernet" of the 1D-CNN of kernel size $(2K' + 1)$ if $K' \leq K$, and the condensed dataset via a bigger model can generalize well to smaller ones. This result suggests us to *use a sufficiently large model during condensation*, to enable the generalization of the condensed dataset to a wider range of models.

**Proposition 3** (Condensed Adjacency Overfits SDC Objective). *Consider least-squares regression with a linear GNN, $f(A, X) = C(A)XW$ parameterized by $W$ and $C(A)$ which depends on graph adjacency $A$. For any (full-ranked) synthetic node features $X' \in \mathbb{R}^{c \times d}$, there exists a synthetic adjacency matrix $A' \in \mathbb{R}_{\geq 0}^{c \times c}$ such that the gradient matching objective is achievable.*

**Proposition 4** (Failed Generalization of SDC across GNNs). *Consider least-squares regression with a linear GNN, $f_W^C(A, X) = C(A)XW$ parametrized by $W$; there always exists a condensed graph $\mathcal{S}^*$, such that the gradient matching objective for $f^C$ is achievable. However, if we train a new linear GNN $f_W^{C'}(A, X)$ with convolution matrix $C'(A')$ on $\mathcal{S}^*$, the relative error between the optimized model parameters of $f_W^{C'}$ on the real and condensed graphs is $\|W_{C'}^{\mathcal{S}} - W_{C'}^{\mathcal{T}}\| / \|W_{C'}^{\mathcal{T}}\| \geq \max\{\sigma_{\max}(Q) - 1, 1 - \sigma_{\min}(Q)\}$, where $W_{C'}^{\mathcal{T}} = \arg\min_W \|\mathbf{y} - f_W^{C'}(A, X)\|_2^2$, $W_{C'}^{\mathcal{S}} = \arg\min_W \|\mathbf{y}' - f_W^{C'}(A', X')\|_2^2$, and $Q = \left( X^\top [C(A)]^\top [C(A)] X \right) \left( X^\top [C'(A)]^\top [C'(A)] X \right)^{-1}$.*

# E   HYPERGRADIENTS AND IMPLICIT FUNCTION THEOREM

In this section, we give some additional background on the concepts of hyperparameter gradients (hypergradient for short) and the implicit function theorem (IFT).

We shall recall the notations we used in the main paper, which are summarized in Table 6.

Table 6: Notations.

| | |
|---:|:---|
| $\lambda, \theta$ | Hyperparameters and NN parameters |
| $\mathcal{T}, \mathcal{S}$ | The original and synthetic datasets |
| $\mathcal{L}_{\mathcal{T}}^{\text{train}}, \mathcal{L}_{\mathcal{T}}^{\text{val}}$ | The training and validation loss on the original dataset |
| $\mathcal{L}_{\mathcal{S}}^{\text{train}}, \mathcal{L}_{\mathcal{S}}^{\text{val}}$ | The training and validation loss on the synthetic dataset |
| $\theta^{\mathcal{T}}(\lambda) := \arg\min_\theta \mathcal{L}_{\mathcal{T}}^{\text{train}}(\theta, \lambda)$ | The optimized parameters on the original dataset, as a function of the hyperparameter |
| $\theta^{\mathcal{S}}(\lambda) := \arg\min_\theta \mathcal{L}_{\mathcal{S}}^{\text{train}}(\theta, \lambda)$ | The optimized parameters on the synthetic dataset, as a function of the hyperparameter |
| $\mathcal{L}_{\mathcal{T}}^*(\lambda) := \mathcal{L}_{\mathcal{T}}^{\text{val}}\left(\theta^{\mathcal{T}}(\lambda), \lambda\right)$ | The validation loss on the original dataset, as a function of the hyperparameter $\lambda$ |
| $\mathcal{L}_{\mathcal{S}}^*(\lambda) := \mathcal{L}_{\mathcal{S}}^{\text{val}}\left(\theta^{\mathcal{S}}(\lambda), \lambda\right)$ | The validation loss on the synthetic dataset, as a function of the hyperparameter $\lambda$ |

In Fig. 7, we visualize the geometrical process to define the hypergradients. First, in (a), we can plot the training loss $\mathcal{L}_{\mathcal{T}}^{\text{train}}(\theta, \lambda)$ as a two-variable function on the parameters $\theta$ and hyperparameter $\lambda$. For each hyperparameter $\lambda \in \Lambda$ (which is assumed to be a continuous interval in this case), we shall optimize the training loss to find an optimal model parameter, which forms the blue curve in $(\theta, \lambda)$-plane in (a). Now, we shall substitute the fitted parameter $\theta^{\mathcal{T}}(\lambda) := \arg\min_\theta \mathcal{L}_{\mathcal{T}}^{\text{train}}(\theta, \lambda)$ as a function of the hyperparameter into the validation loss $\mathcal{L}_{\mathcal{T}}^{\text{val}}(\theta, \lambda)$. Geometrically, it is projecting the blue implicit optimal parameter curve onto the surface of the validation loss, as shown in (b). The projected orange curve is the validation loss as a one-variable function of the hyperparameter lambda $\mathcal{L}_{\mathcal{T}}^*(\lambda) := \mathcal{L}_{\mathcal{T}}^{\text{val}}\left(\theta^{\mathcal{T}}(\lambda), \lambda\right)$. Finally, the purple curve represents the hyperparameter gradients, which is the slope of the tangent line on the orange validation loss curve.

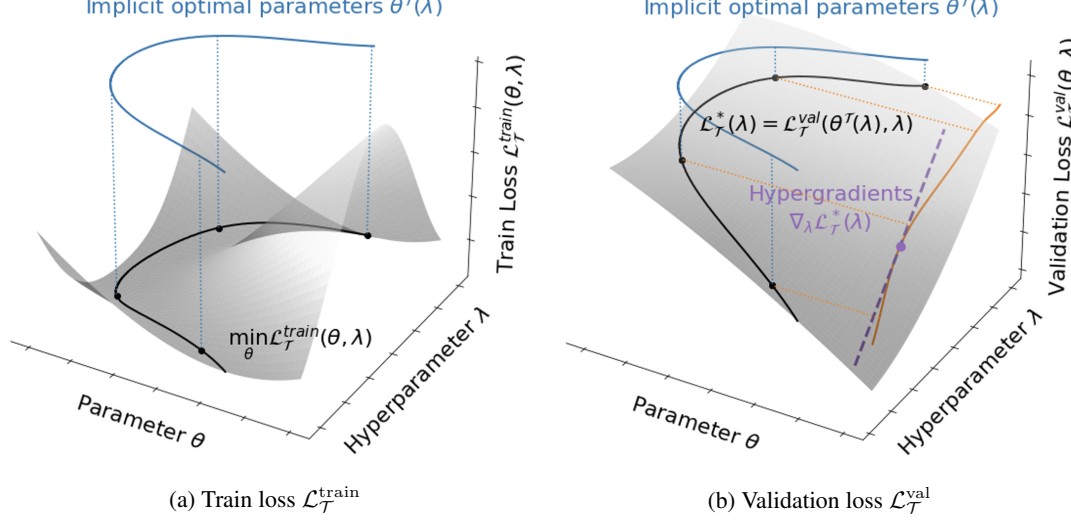

(a) Train loss $\mathcal{L}_{\mathcal{T}}^{\text{train}}$   (b) Validation loss $\mathcal{L}_{\mathcal{T}}^{\text{val}}$

Figure 7: Loss landscape w.r.t. $\theta$ and $\lambda$. A hyperparameter $\lambda$ has an optimal parameter $\theta^{\mathcal{T}}(\lambda)$ (blue curve in $(\theta, \lambda)$-plane in (**a**)) that minimizes the train loss. In (**b**), injecting optimal parameters $\theta^{\mathcal{T}}(\lambda)$ into the validation loss, we obtain a function of validation loss w.r.t. $\lambda$ (denoted as $\mathcal{L}_{\mathcal{T}}^\star(\lambda)$) in $(\mathcal{L}, \lambda)$-plane, shown as the orange curve. The purple dash line illustrates the hypergradients, i.e., gradient of $\mathcal{L}_{\mathcal{T}}^\star(\lambda)$ w.r.t. $\lambda$.

# F   MORE THEORETICAL RESULTS

In this section, we provide the proofs to the theoretical results Lemma 5 and Propositions 2 to 4 and Theorem 1, together with some extended theoretical discussions, including generalizing the *linear convolution regression problem* to non-convex losses and non-linear models (see Appendix F.4).

To proceed, please recall the *linear convolution regression problem* defined in Appendix C.5, the achievability of gradient-matching objective (Eq. (2)) defined as Assumption 1 in Appendix D.

### F.1 VALIDITY OF STANDARD DATASET CONDENSATION

As the first step, we verify the validity of the standard dataset condensation (SDC) using the gradient-matching objective Eq. (2) for the *linear convolution regression problem*.

**Lemma 5.** *(Validity of SDC) Consider least square regression with linear convolution model $f_W(A, X) = C(A)XW$ parameterized by $W$. If the gradient-matching objective of $f_W$ is achievable, then the optimizer on the condensed dataset $\mathcal{S}$, i.e., $W^{\mathcal{S}} = \arg\min_W \mathcal{L}_{\mathcal{S}}(W)$ is also optimal for the original dataset, i.e., $\mathcal{L}_{\mathcal{T}}(W^{\mathcal{S}}) = \min_W \mathcal{L}_{\mathcal{T}}(W)$.*

*Proof.* In the *linear convolution regression problem*, sum-of-squares loss is used, and $\mathcal{L}_{\mathcal{T}}(W) = \|CXW - \mathbf{y}\|_2^2$ (similarly $\mathcal{L}_{\mathcal{S}}(W) = \|C'X'W - \mathbf{y}'\|_2^2$ where $C' = C(A')$). We assume $X^\top C^\top CX \in \mathbb{R}^{d \times d}$ is invertible and we can apply the optimizer formula for ordinary least square (OLS) regression to find the optimizer $W^{\mathcal{T}}$ of $\mathcal{L}_{\mathcal{T}}(W)$ as,

$$W^{\mathcal{T}} = (X^\top C^\top CX)^{-1}X^\top C^\top \mathbf{y}.$$

Also, we can compute the gradients of $\mathcal{L}_{\mathcal{T}}(W)$ w.r.t $W$ as,

$$\nabla_W \mathcal{L}_{\mathcal{T}}(W) = 2X^\top C^\top (CXW - \mathbf{y}),$$

and similarly for $\nabla_W \mathcal{L}_{\mathcal{S}}(W)$.

Given the achievability of the gradient-matching objective of $f_W$, we know there exists a non-degenerate trajectory $(W_t^{\mathcal{S}})_{t=0}^{T-1}$ which spans the entire parameter space, i.e., $\text{span}(W_0^{\mathcal{S}}, \ldots, W_{T-1}^{\mathcal{S}}) = \mathbb{R}^d$, such that the gradient-matching loss (the objective of Eq. (2) without expectation) on this trajectory is 0. Assuming $D(\cdot, \cdot)$ is the $L_2$ norm (Zhao et al., 2020), this means,

$$\nabla_W \mathcal{L}_{\mathcal{T}}(W_t^{\mathcal{S}}) = \nabla_W \mathcal{L}_{\mathcal{S}}(W_t^{\mathcal{S}}) \quad \text{for} \quad t \in [T].$$

Substitute in the formula for the gradients $\nabla_W \mathcal{L}_{\mathcal{T}}(W)$ and $\nabla_W \mathcal{L}_{\mathcal{S}}(W)$, we then have,

$$X^\top C^\top (CXW_t^{\mathcal{S}} - \mathbf{y}) = X'^\top C'^\top (C'X'W_t^{\mathcal{S}} - \mathbf{y}') \quad \text{for} \quad t \in [T].$$

Since the set of $\{W_t^{\mathcal{S}}\}_{t=0}^{T-1}$ spans the complete parameter space $\mathbb{R}^d$, we can transform the set of vectors $\{\omega_t \cdot W_t^{\mathcal{S}}\}_{t=0}^{T-1}$ to the set of unit vectors $\{\mathbf{e}_i^d\}_{i=0}^{d-1} \in \mathbb{R}^d$ by a linear transformation. Meanwhile, the set of $T$ equations above can be transformed to,

$$X^\top C^\top (CX\mathbf{e}_i^d - \mathbf{y}) = X'^\top C'^\top (C'X'\mathbf{e}_i^d - \mathbf{y}') \quad \text{for} \quad i \in [d].$$

This directly leads to $X^\top C^\top CX = X'^\top C'^\top C'X'$ and $X^\top C^\top \mathbf{y} = X'^\top C'^\top \mathbf{y}'$.

Using the formula for the optimizers $W^{\mathcal{T}}$ and $W^{\mathcal{S}}$ above, we readily get,

$$W^{\mathcal{T}} = (X^\top C^\top CX)^{-1}X^\top C^\top \mathbf{y} = (X'^\top C'^\top C'X')^{-1}X'^\top C'^\top \mathbf{y}' = W^{\mathcal{S}}.$$

And hence,

$$\mathcal{L}_{\mathcal{T}}(W^{\mathcal{S}}) = \mathcal{L}_{\mathcal{T}}(W^{\mathcal{T}}) = \min_W \mathcal{L}_{\mathcal{T}}(W),$$

which concludes the proof. $\qquad\square$

Despite its simplicity, Lemma 5 directly verifies the validity of the gradient-matching formulation of standard dataset condensation on some specific learning problems. Although the gradient-matching formulation (Eq. (2)) is an efficient but weaker formulation than the bilevel formulation of SDC (Eq. (SDC)), we see it is strong enough for some of the *linear convolution regression problem*.

### F.2 GENERALIZATION ISSUES OF SDC

Now, we move forward and focus on the generalization issues of (the gradient-matching formulation of) the standard dataset condensation (SDC) across GNNs.

First, we prove the successful generalization of SDC across 1D-CNNs as follows, which is very similar to the proof of Lemma 5.

*Proof of Proposition 2:* In Proposition 2, we consider one-dimensional linear convolution models $f^{2K+1}(X) = (\sum_{k=-K}^{k=K} \alpha^{(k)} P^k) XW$ parameterized by $\alpha \in \mathbb{R}^p$ and $W \in \mathbb{R}^d$ (where $p = 2K + 1$). If we denote,

$$C = \sum_{k=-K}^{k=K} \alpha^{(k)} P^k \quad \text{and} \quad \theta = [\alpha, W] \in \mathbb{R}^{p+d}$$

then from the proof of Lemma 5 we know the gradients of $\mathcal{L}_{\mathcal{T}}(W)$ w.r.t $W$ is again,

$$\nabla_W \mathcal{L}_{\mathcal{T}}(W) = 2X^\top C^\top (CXW - \mathbf{y}).$$

We know the achievability of the gradient-matching objective means there exists a non-degenerate trajectory $(\theta_t^{\mathcal{S}})_{t=0}^{T-1}$ which spans the entire parameter space, i.e., $\text{span}(\theta_0^{\mathcal{S}}, \dots, \theta_{T-1}^{\mathcal{S}}) = \mathbb{R}^{p+d}$. By decomposing $\theta_t^{\mathcal{S}}$ into $[\alpha_t^{\mathcal{S}}, W_t^{\mathcal{S}}]$, we know that there exists $(\alpha_t^{\mathcal{S}})_{t=0}^{T-1}$ which spans $\mathbb{R}^p$ and there exists $(W_t^{\mathcal{S}})_{t=0}^{T-1}$ which spans $\mathbb{R}^d$.

Since the gradient-matching objective is minimized to 0 on $(W_t^{\mathcal{S}})_{t=0}^{T-1}$ which spans $\mathbb{R}^d$, following the same procedure as the proof of Lemma 5, we again obtain,

$$X^\top C^\top \mathbf{y} = X'^\top C'^\top \mathbf{y}'.$$

Meanwhile, since the same gradient-matching objective is also minimized to 0 on $(\alpha_t^{\mathcal{S}})_{t=0}^{T-1}$ which spans $\mathbb{R}^p$, we have,

$$X^\top \Big( \sum_{k=-K}^{k=K} (\alpha_t^{\mathcal{S}})^{(k)} P^k \Big)^\top \mathbf{y} = X'^\top \Big( \sum_{k=-K}^{k=K} (\alpha_t^{\mathcal{S}})^{(k)} P'^k \Big)^\top \mathbf{y}' \quad \text{for} \quad t \in [T].$$

Again by linear combining the above $T$ equations and because $(\alpha_t^{\mathcal{S}})_{t=0}^{T-1}$ can be transformed to the unit vectors in $\mathbb{R}^p$, we have,

$$X^\top (P^k)^\top \mathbf{y} = X'^\top (P'^k)^\top \mathbf{y}' \quad \text{for} \quad k = -K, \dots, K.$$

Hence, for any new trajectory $(\alpha_t'^{\mathcal{S}})_{t=0}^{T-1}$ which spans $\mathbb{R}^{p'}$ where $p' = 2K' + 1$, by linear combining the above equations, we have,

$$X^\top \Big( \sum_{k=-K}^{k=K} (\alpha_t'^{\mathcal{S}})^{(k)} P^k \Big)^\top \mathbf{y} = X'^\top \Big( \sum_{k=-K}^{k=K} (\alpha_t'^{\mathcal{S}})^{(k)} P'^k \Big)^\top \mathbf{y}' \quad \text{for} \quad t \in [T'].$$

With similar procedure for the $X^\top C^\top CX$ part, we conclude that on the new trajectory $(\theta_t'^{\mathcal{S}})_{t=0}^{T-1}$

$$\nabla_W \mathcal{L}_{\mathcal{T}}(\alpha, W) = \nabla_W \mathcal{L}_{\mathcal{S}}(\alpha, W).$$

It remains to prove that on any new trajectory $\nabla_\alpha \mathcal{L}_{\mathcal{T}}(\alpha, W) = \nabla_\alpha \mathcal{L}_{\mathcal{S}}(\alpha, W)$. Only need to note that,

$$\nabla_{\alpha^{(k)}} \mathcal{L}_{\mathcal{T}}(\alpha, W) = 2W^\top X^\top P^k (CXW - \mathbf{y}).$$

Hence, by the $p$ equations above, we can readily show,

$$X^\top P^k \mathbf{y} = X'^\top P'^k \mathbf{y}' \quad \text{for} \quad k = -K, \dots, K.$$

Again with a similar procedure for the $X^\top C^\top CX$ part, we finally can show that on the new trajectory $(\theta_t'^{\mathcal{S}})_{t=0}^{T-1}$

$$\nabla_\alpha \mathcal{L}_{\mathcal{T}}(\alpha, W) = \nabla_\alpha \mathcal{L}_{\mathcal{S}}(\alpha, W).$$

This concludes the proof. □

Then we focus on the linear GNNs, we want to verify the insight that the learned adjacency $A'$ of the condensed graph has "too many degrees of freedom" so that can easily overfit the gradient-matching objective, no matter what learned synthetic features $X'$ are. Again, the proof of Proposition 3 uses some results in the proof of Lemma 5.

*Proof of Proposition 3:* Now, we consider a linear GNN defined as $f(A, X) = C(A)XW$. From the proof of Lemma 5, we know that for the gradient-matching objective of $f$ to be achievable, it is equivalent to require that,

$$X^\top C^\top CX = X'^\top C'^\top C'X' \quad \text{and} \quad X^\top C^\top \mathbf{y} = X'^\top C'^\top \mathbf{y}',$$

where $C$ and $C'$ refer to $C(A)$ and $C(A')$ respectively.

Firstly we note that once we find $C'$ and $X'$ such that satisfy the first condition $X^\top C^\top CX = X'^\top C'^\top C'X'$, we can always find $\mathbf{y}' \in \mathbb{R}^c$ such that $X^\top C^\top \mathbf{y} = X'^\top C'^\top \mathbf{y}'$ since $X^\top C^\top \mathbf{y} \in \mathbb{R}$ is a scalar.

Now, we focus on finding the convolution matrix $C'$ and the node feature matrix $X'$ of the condensed synthetic graph to satisfy $X^\top C^\top CX = X'^\top C'^\top C'X'$. We assume $n \gg c \gg d$ and consider the diagonalization of $X^\top C^\top CX \in \mathbb{R}^{d \times d}$. Since $X^\top C^\top CX$ is positive semi-definite, it can be diagonalized as $X^\top C^\top CX = VS^2V^\top$ where $V \in \mathbb{R}^d$ is an orthogonal matrix and $S \in \mathbb{R}^d$ is a diagonal matrix that $S = \text{diag}(s_1, \ldots, s_d)$.

For any (real) semi-unitary matrix $U \in \mathbb{R}^{c \times d}$ such that $U^\top U = I_d$, we can construct $C'X' = USV^\top \in \mathbb{R}^{c \times d}$ and we can easily verify they satisfy the condition,

$$X'^\top C'^\top C'X' = VSU^\top USV^\top = VS^2V^\top = X^\top C^\top CX.$$

Then since $X'$ is full ranked, for any $X'$, by considering the singular-value decomposition of $X'$, we see that we can always find a convolution matrix $C'$ such that $C'X' = USV$ and this concludes the proof. □

Finally, we use some results of Proposition 3 to prove Proposition 4, the failure of SDC when generalizing across GNNs.

*Proof of Proposition 4:* We prove this by two steps.

For the first step, we aim to show that there always exist a condensed synthetic dataset $\mathcal{S}$ such that achieves the gradient-matching objective but the learned adjacency matrix $A' = I_c$ is the identity matrix. Clearly this directly follows form the proof of Proposition 3, where we only require $C'X' = USV$ (see the proof of Proposition 3 for details). If the learned adjacency matrix $A' = I_c$, the for any GNNs, the corresponding convolution matrix $C'$ is also (or proportional to) identity, thus we only need to set the learned node feature matrix $X' = USV$ to satisfy the condition. The first step is proved.

For the second step, we evaluate the relative estimation error of the optimal parameter when transferred to a new GNN $f_W^{\mathscr{C}}$ with convolution filter $\mathscr{C}(\cdot)$, i.e., $\|W_{\mathscr{C}}^{\mathcal{S}} - W_{\mathscr{C}}^{\mathcal{T}}\| / \|W_{\mathscr{C}}^{\mathcal{T}}\|$. Using the formula for the optimal parameter in the proof of Lemma 5 again, we have,

$$W_{\mathscr{C}}^{\mathcal{T}} = (X^\top \mathscr{C}^\top \mathscr{C}X)^{-1} X^\top \mathscr{C}^\top \mathbf{y},$$

and

$$W_{\mathscr{C}}^{\mathcal{S}} = (X'^\top \mathscr{C}'^\top \mathscr{C}'X)^{-1} X'^\top \mathscr{C}'^\top \mathbf{y}',$$

where $\mathscr{C}' = \mathscr{C}(A') = \mathscr{C}(I_c) = C(I_c)$ (the last equation use the fact that the convolution matrix of GNNs are the same if the underlying graph is identity).

Moreover, by the validity of SDC on $f_W^C$, we know, (see the proof of Lemma 5 for details),

$$X'^\top C'^\top C'X' = X^\top C^\top CX \quad \text{and} \quad X'^\top C'^\top \mathbf{y}' = X^\top C^\top \mathbf{y}$$

Thus, altogether we derive that $X'^\top \mathscr{C}'^\top \mathscr{C}'X = X^\top C^\top CX$ and $X'^\top \mathscr{C}'^\top \mathbf{y}' = X^\top C^\top \mathbf{y}$. And therefore,

$$W_{\mathscr{C}}^{\mathcal{S}} = (X^\top C^\top CX)^{-1} X^\top C^\top \mathbf{y}.$$

Now, note that,

$$
\begin{aligned}
&\|W_{\mathscr{C}}^{\mathcal{S}} - W_{\mathscr{C}}^{\mathcal{T}}\| / \|W_{\mathscr{C}}^{\mathcal{T}}\| \\
&= \left\| \left( \left( X^\top [C(A)]^\top [C(A)] X \right) \left( X^\top [\mathscr{C}(A)]^\top [\mathscr{C}(A)] X \right)^{-1} \right) - I_d \right) X^\top C^\top \mathbf{y} \right\| / \|X^\top C^\top \mathbf{y}\| \\
&\geq \max\{\sigma_{\max}(Q) - 1, 1 - \sigma_{\min}(Q)\}
\end{aligned}
$$

where $Q = \left( X^\top [C(A)]^\top [C(A)] X \right) \left( X^\top [\mathscr{C}(A)]^\top [\mathscr{C}(A)] X \right)^{-1}$. This concludes the proof. $\qquad\square$

## F.3   Validity of HCDC

Finally, we complete the proof of Theorem 1 with more detials.

*Proof of Theorem 1:* Firstly, we prove the *necessity* by contradiction.

If there exists $\lambda_0 \in \tilde{\Lambda}$ s.t. the two gradient vectors are not aligned at $\lambda_0$, then there exists small perturbation $\Delta\lambda_0$ such that $\mathcal{L}_{\mathcal{T}}^*(\lambda_0 + \Delta\lambda_0) - \mathcal{L}_{\mathcal{T}}^*(\lambda_0)$ and $\mathcal{L}_{\mathcal{S}}^*(\lambda_0 + \Delta\lambda_0) - \mathcal{L}_{\mathcal{S}}^*(\lambda_0)$ have different signs.

Secondly, we prove the *sufficiency* by path-integration.

For any pair $\lambda_1 \neq \lambda_2 \in \tilde{\Lambda}$, we have a path $\gamma(\lambda_1, \lambda_2) \in \tilde{\Lambda}$ from $\lambda_2$ and $\lambda_1$, then integrating hypergradients $\nabla_\lambda \mathcal{L}_{\mathcal{T}}^*(\lambda)$ along the path recovers the hyperparameter-calibration condition. More specifically, along the path we have $\mathcal{L}_{\mathcal{T}}^*(\lambda_1) - \mathcal{L}_{\mathcal{T}}^*(\lambda_2) = \int_{\gamma(\lambda_1, \lambda_2)} \nabla_\lambda \mathcal{L}_{\mathcal{T}}^*(\lambda) \mathrm{d}\lambda$ (similar for $\nabla_\lambda \mathcal{L}_{\mathcal{S}}^*(\lambda)$). Thus we have,

$$
\begin{aligned}
&(\mathcal{L}_{\mathcal{T}}^*(\lambda_1) - \mathcal{L}_{\mathcal{T}}^*(\lambda_2))(\mathcal{L}_{\mathcal{S}}^*(\lambda_1) - \mathcal{L}_{\mathcal{S}}^*(\lambda_2)) \\
&= \left( \int_{\gamma(\lambda_1, \lambda_2)} \nabla_\lambda \mathcal{L}_{\mathcal{T}}^*(\lambda) \mathrm{d}\lambda \right) \left( \int_{\gamma(\lambda_1, \lambda_2)} \nabla_\lambda \mathcal{L}_{\mathcal{S}}^*(\lambda) \mathrm{d}\lambda \right) \\
&\geq \int_{\gamma(\lambda_1, \lambda_2)} \left\langle \sqrt{\nabla_\lambda \mathcal{L}_{\mathcal{T}}^*(\lambda)}, \sqrt{\nabla_\lambda \mathcal{L}_{\mathcal{S}}^*(\lambda)} \right\rangle \mathrm{d}\lambda \\
&\geq 0
\end{aligned}
$$

the second last inequality by Cauchy-Schwarz inquality and the last inequality by $\cos(\nabla_\lambda \mathcal{L}_{\mathcal{T}}^*(\lambda), \nabla_\lambda \mathcal{L}_{\mathcal{T}}^*(\lambda)) = 0$ for any $\lambda \in \gamma(\lambda_1, \lambda_2) \in \tilde{\Lambda}$.

This concludes the proof. $\qquad\square$

## F.4   Generalize to Non-Convex and Non-Linear Case

Although the results above are obtained for least squares loss and linear convolution model, it *still reflects the nature of general non-convex losses and non-linear models*. Since dataset condensation is effectively matching the local minima $\{\theta^{\mathcal{T}}\}$ of the original loss $\mathcal{L}_{\mathcal{T}}^{train}(\theta, \psi)$ with the local minima $\{\theta^{\mathcal{S}}\}$ of the condensed loss $\mathcal{L}_{\mathcal{S}}^{train}(\theta, \psi)$, within the small neighborhoods surrounding the pair of local minima $(\theta^{\mathcal{T}}, \theta^{\mathcal{S}})$, we can approximate the non-convex loss and non-linear model with a convex/linear one respectively. Hence the generalizability issues with convex loss and liner model may hold.

## G   HCDC on Discrete and Continuous Search Spaces

In this subsection, we illustrate how to tackle the two types of search spaces: (1) discrete and finite $\Lambda$ and (2) continuous and bounded $\Lambda$, respectively, illustrated by the practical search spaces of ResNets on images and Graph Neural Networks (GNNs) on graphs.

**Discrete search space of neural architectures on images.**     Typically the neural architecture search (NAS) problem aims to find the optimal neural network architecture with the best validation performance on a dataset from a large set of candidate architectures. One may simply train the set of $p$ pre-defined architectures $\{f^{(i)} \mid i = 1, \ldots, p\}$ and rank their validation performance. We can transform this problem as a continuous HPO, by defining an "interpolated" model (i.e., a supernet (Wang et al., 2020)) $f_\theta^\lambda$, where hyperparameters $\lambda = [\lambda^{(1)}, \ldots, \lambda^{(p)}] \in \Lambda$ and $\theta$ is the model

parameters. This technique is known as the differentiable NAS approach (see Appendix B), e.g., DARTS (Liu et al., 2018), which usually follows a cell-based search space (Zoph et al., 2018) and continuously relaxes the original discrete search space. Subsequently, in (Xie et al., 2018; Dong & Yang, 2019), the Gumbel-softmax trick (Jang et al., 2017; Maddison et al., 2017) is applied to approximate the hyperparameter gradients. We apply the optimization strategy in GDAS (Dong & Yang, 2019), which also provides the approximations of the hypergradients in our experiments.

**Continuous search space of graph convolution filters.** Many graph neural networks (GNNs) can be interpreted as performing message passing on node features, followed by feature transformation and an activation function. In this regard, these GNNs differ from each other by choice of convolution matrix (see Appendix D for details). We consider the problem of searching for the best convolution filter of GNNs on a graph dataset. In Appendices D and F, we theoretically justify that dataset condensation for HPO is challenging on graphs due to overfitting issues. Nonetheless, this obstacle is solved by HCDC.

A natural continuous search space of convolution filters often exists in GNNs, e.g., when the candidate convolution filters can be expressed by a generic formula. For example, we make use of a truncated powers series of the graph Laplacian matrix to model a wide range of convolution filters, as considered in ChebNet (Defferrard et al., 2016) or SIGN (Frasca et al., 2020). Given the differentiable generic formula of the convolution filters, we can treat the convolution filter (or, more specifically, the parameters in the generic formula) as hyperparameters and evaluate the hypergradients using the implicit differentiation methods discussed in Section 5.1.

## H    EFFICIENT DESIGN OF HCDC ALGORITHM

For continuous hyperparameters, $\Lambda$ itself is usually compact and connected, and the minimal extended search space is $\tilde{\Lambda} = \Lambda$. Consider a discrete search space $\Lambda$, which consists of $p$ candidate hyperparameters. We can naively construct $\tilde{\Lambda}$ as $O(p^2)$ continuous paths connecting pairs of candidate hyperparameters (see Fig. 8a in Appendix H for illustration). This is undesirable due to its quadratic complexity in $p$; for example, the number of candidate architectures $p$ is often a large number for practical NAS problems.

We propose a construction of $\tilde{\Lambda}$ with linear complexity in $p$, which works as follows. For any $i \in [p]$, we construct a "representative" path, named $i$-th HPO trajectory, which starts from $\lambda_{i,0}^{\mathcal{S}} = \lambda_i \in \Lambda$ and updates through $\lambda_{i,t+1}^{\mathcal{S}} \leftarrow \lambda_{i,t}^{\mathcal{S}} - \eta \nabla_\lambda \mathcal{L}_{\mathcal{S}}^*(\lambda_{i,t}^{\mathcal{S}})$, shown as the orange dashed lines in Fig. 8a. We assume all of the $p$ trajectories will approach the same or equivalent optima $\lambda^{\mathcal{S}}$, forming "connected" paths (i.e., orange dashed lines which merge at the optima $\lambda^{\mathcal{S}}$) between any pair of hyperparameters $\lambda_i \neq \lambda_j \in \Lambda$. This construction is also used in a continuous search space (as shown in Fig. 8b) to save computation (except that we have to select the starting randomly points $\lambda_i \sim \mathbb{P}_\Lambda$).

## I    COMPLEXITY ANALYSIS OF HCDC ALGORITHM

In this section, we provide additional details on the complexity analysis of the HCDC algorithm Algorithm 1. Following the algorithm pseudocode, we consider the discrete hyperparameter search space of size $p$. The overall time complexity of HCDC is proportional to this size $p$ of the hyperparameter search space.

If we denote the dimensionality of the model parameters $\theta$ and hyperparameters $\lambda$ by $P$ and $H$ respectively. We know the time complexity of the common model parameter update is $O(P)$. Based on (Lorraine et al., 2020), the hyperparameter update Line 7 needs $O(P + H)$ time (since we fixed the truncated power of the Neumann series approximation as constant) and $O(P + H)$ memory. In Line 8, we need another backpropagation to take gradients of $\nabla_\lambda \mathcal{L}_{\mathcal{S}}^{\text{val}}(\theta, \lambda)$ w.r.t. $\mathcal{S}^{\text{val}}$. This update is performed in a mini-batch manner and supposes the dimensionality of validation samples in the mini-batch is $B$, the Line 8 requries $O(HB)$ time and memory.

## J    IMPLEMENTATION DETAILS

In this section, we list more implementation details on the experiments in Section 7.

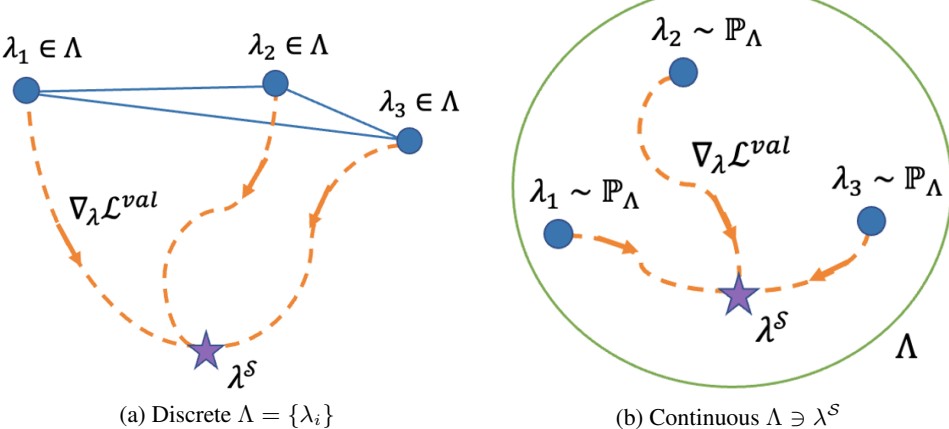

(a) Discrete $\Lambda = \{\lambda_i\}$             (b) Continuous $\Lambda \ni \lambda^{\mathcal{S}}$

Figure 8: Illustration of the constructed extended search space $\tilde{\Lambda}$ illustrated as the orange trajectory for both **a)** discrete $\Lambda$ and **(b)** continuous $\Lambda$. The trajectory starts from $\lambda_{i,0}^{\mathcal{S}} = \lambda_i \in \Lambda$ for discrete $\Lambda$ (or random points for continuous $\Lambda$), and updates through $\lambda_{i,t+1}^{\mathcal{S}} \leftarrow \lambda_{i,t}^{\mathcal{S}} - \eta \nabla_\lambda \mathcal{L}_{\mathcal{S}}^*(\lambda_{i,t}^{\mathcal{S}})$.

For the synthetic experiments on CIFAR-10, we randomly split the CIFAR-10 images into $M = 20$ splits and perform cross-validation. For the baseline methods (Random, SDC-GM, SDC-DM), the dataset condensation is performed independently for each split. For HCDC, we first condense the training set of the synthetic dataset by SDC-GM. Then, we learn a separate validation set with $1/M$-size of the training set and train with the HCDC objective on the $M$-HPO trajectories as described in Section 5. We report the correlation between the ranking of splits (in terms of their validation performance on this split). For the Early-Stopping method, we only train the same number of iterations as the other methods (with the same batchsize), which means there are only $c/n * 500$ epochs.

For the experiments about finding the best convolution filter on (large) graphs, we create the set of ten candidate convolution filters as (see Table 4 for definitions and references) GCN, SAGE-Mean, SAGE-Max, GAT, GIN-$\epsilon$, GIN-0, SGC(K=2), SGC(K=3), ChebNet(K=2), ChebNet(K-3). The implementations are provided by PyTorch Geometric https://pytorch-geometric.readthedocs.io/en/latest/modules/nn.html. We also select the GNN width from $\{128, 256\}$ and the GNN depth from $\{2, 4\}$ so there are $10 \times 2 \times 2 = 40$ models in total.

For the experiments about speeding up off-the-shelf graph architecture search algorithms, we adopt GraphNAS (Gao et al., 2019) together with their proposed search space from their official repository https://github.com/GraphNAS/GraphNAS. We apply to the ogbn-arxiv graph with condensation ratio $c_{train}/n = 0.5\%$.

## K   MORE EXPERIMENTS

**Synthetic experiments on CIFAR-10.** We first consider a synthetically created set of hyperparameters on the image dataset, CIFAR-10. Consider the $M$-fold cross-validation, where a fraction of $1/M$ samples are used as the validation split each time. The $M$-fold cross-validation process can be modeled by a set of $M$ hyperparameters $\{\varphi_i \in \{0, 1\} \mid i = 1, \ldots, M\}$, where $\varphi_i = 1$ if and only if the $i$-th fold is used for validation. The problem of finding the best validation performance among the $M$ results can be modeled as a hyperparameter optimization problem with a discrete search space $|\Psi| = M$. We compare HCDC with the gradient-matching (Zhao et al., 2020) and distribution matching (Zhao & Bilen, 2021b) baselines. We also consider a uniform random sampling baseline and an early-stopping baseline where we train only $c/n * 500$ epochs but on the original dataset. The results of $M = 20$ and $c/n = 1\%$ is reported in Table 7, where we see HCDC achieves the highest rank correlation.

Table 7: The rank correlation and validation performance on the original dataset of the $M$-fold cross-validation ranked/selected on the condensed dataset on CIFAR-10.

| Method | Ratio ($c_{\mathrm{train}}/n$) | |
| --- | --- | --- |
| | 2% | 4% |
| Random | $-0.03$ | 0.07 |
| SDC-GM | 0.64 | 0.78 |
| SDC-DM | 0.77 | 0.86 |
| Early-Stopping | 0.11 | 0.24 |
| HCDC | 0.91 | 0.94 |