# OpenReview forum: "Calibrated Dataset Condensation for Faster Hyperparameter Search"
_ICLR.cc/2024/Conference — Submitted to ICLR 2024_

### Official Review · Reviewer_EsH1 · 2023-10-18

**Soundness:** 4 excellent
**Presentation:** 4 excellent
**Contribution:** 4 excellent
**Rating:** 6
**Confidence:** 5

**Summary:**

This paper proposes a dataset condensation/distillation method specifically for the application of hyperparameter search. The authors first demonstrate the drawback of existing dataset condensation methods in this area and analyze the inefficiency of a naive solution that involves solving nested optimization. Then, through theoretical analysis, they propose a hyperparameter-calibrated dataset condensation method. The basic idea is to learn a synthetic validation dataset such that hypergradients on hyperparameters can be matched. They also apply effective approximation to further reduce the complexity so that the final algorithm is a first-order one. Experiments validate the effectiveness of the proposed method on both image and graph datasets.

**Strengths:**

* The research on a dataset condensation method specifically for hyperparameter search is meaningful. One important application of dataset condensation/distillation is to boost the training efficiency of neural network to facilitate researches like NAS. However, as demonstrated by the authors, existing methods fail to do so both theoretically and experimentally.
* The presentation is really good and coherent generally. The authors first demonstrate the drawbacks of existing methods and a naive solution before presenting the final solution, which indicates that the proposed method is well-motivated, reasonable, and interesting.

**Weaknesses:**

1. Limited evaluation. The experiments are only conducted on small-scale datasets like CIFAR, Cora, Citeseer, Ogbn-arxiv, and Reddit. To better demonstrate the scalability and robustness of the proposed method, experiments on larger datasets like ImageNet, at least subsets of ImageNet or TinyImageNet, are encouraged.
2. Limited Analysis. The authors only focus on the performance on several datasets. The analysis on the algorithm itself is insufficient. Specifically, how does the algorithm compare to the naive solution in Eq. 3 in terms of both performance and efficiency. The authors may consider using a toy dataset for this analysis. After all, the proposed method is an efficient approximation of directly using original datasets. Where the approximation comes from and how much it is are important in a scientific view.
3. I am curious about the details of the method to deal with discrete hyperparameters. The descriptions in Sec. 5.2 and Appendix H are confusing to me. An algorithmic flow is encouraged for better presentation. In addition, are other approaches like Gumbel Softmax applicable?
4. Eq. IFT is an efficient approximation of the hypergradients. It would be better to make this clear in the description. And "=" should not be used in Eq. IFT.
5. Minor: some confusing expersions:
   * "this limitation" in the second paragraph of Introduction: the contents in the previous paragraph is not actually a limitation. And this sentence is incoherent to the last sentence of the previous paragraph.
   * The last two lines in Page 5.
   * Page 6: "For generating $S^{train}$, We ...", "W" here should be lowercase.

**Questions:**

Please refer to Weaknesses.

---

> ### Author Response · Authors · 2023-11-22
> **Response to Reviewer EsH1 (1/2)**
>
> We thank reviewer EsH1, for their service and constructive feedback. We appreciate that the reviewer recognizes the importance of the topic and the strengths of the paper. In particular, we are encouraged that reviewer EsH1 finds that our proposed problem is important and novel, our presentation is really good and coherent generally, our demonstration of the drawbacks of existing methods is significant, and our method is well-motivated, reasonable, and interesting.
>
> Below we address the concerns from reviewer EsH1.
>
>
>
> > **Weakness 1:** Limited evaluation. The experiments are only conducted on small-scale datasets like CIFAR, Cora, Citeseer, Ogbn-arxiv, and Reddit. To better demonstrate the scalability and robustness of the proposed method, experiments on larger datasets like ImageNet, at least subsets of ImageNet or TinyImageNet, are encouraged.
>
> **Response to Weakness 1:**
>
> - Our experimental focus of the paper is on the CIFAR-10 and CIFAR-100 datasets. We acknowledge the significance of testing on larger datasets like ImageNet. However, conducting Neural Architecture Search (NAS) experiments on the scale of ImageNet would be extremely time-consuming, making it hard to conduct in an academic setup. The efficacy and scalability of our dataset condensation approach are demonstrated through our successful application on CIFAR-10 and CIFAR-100, as well as large scale graph data. The main goal of our paper is to introduce an innovative approach to the community, as opposed to setting state-of-the-art NAS results on vast image classification datasets.
>
> - We thank the reviewer for this suggestion. We are in the process of evaluating our method on a large architecture. However, we would like to emphasize that
>     1. We have conducted the architecture search over randomly sampled 100 network models from 15,625 networks for CIFAR-10 and CIFAR-100. Our experiments compare with 7 baselines. All experiments are repeated over 5 random seeds. So overall, we have worked over # of NETWORKS $\times$ (# of baselines +1) $\times$ random seeds = 100 $\times$ 8 $\times$ 5 = 4000 models. We believe this is strong enough evidence that our methods outperform others with substantial statistical significance.
>     2. We have not only verified the performance on vision tasks but also have included hyperparameter search over Graph Neural Networks. Experimentally, we have shown four graph datasets, including two large graphs with more than 100K nodes.
>     3. Our method is combined with off-the-shelf NAS algorithms. 100 models are used here for this experiment as well.
>
> - A recent benchmarking study, "DC-BENCH: Dataset Condensation Benchmark," published at NeurIPS 2022 (https://arxiv.org/pdf/2207.09639.pdf), conducted experiments on CIFAR 10 and CIFAR 100 too. Consistent with this, our paper also includes experiments on CIFAR 10 and CIFAR 100, but additionally, we expand our investigation to include large-scale graph data.
>
> - We plan to address this experiment in future work, aiming to significantly enhance the method's efficiency and scalability. Despite these challenges, we believe our work merits publication due to its innovative approach and being the first to tackle the issue of hyperparameter search via data condensation.

---

> > ### Author Response · Authors · 2023-11-22
> > **Response to Reviewer EsH1 (2/2)**
> >
> > > Weakness 2: Limited Analysis. The authors only focus on the performance on several datasets. The analysis on the algorithm itself is insufficient. Specifically, how does the algorithm compare to the naive solution in Eq. 3 in terms of both performance and efficiency. The authors may consider using a toy dataset for this analysis. After all, the proposed method is an efficient approximation of directly using original datasets. Where the approximation comes from and how much it is are important in a scientific view.
> >
> > **Response to Weakness 2:**
> > - We appreciate reviewer's valuable suggestion for an ablation study on comparing the first-order approximation method we deployed (hypergradient matching) to the original bilevel optimzation problem in Eq. 3. We agree that conducting such a study would further solidify our findings.
> >
> >
> > - However, due to space constraints, we prioritized presenting the core message of this paper: data condensation that preserves hyperparameter performance for hyperparameter search. To effectively convey this message, we conducted extensive experiments. As a result, we were unable to include an extensive ablation study on the first-order approximation method.
> >
> > - It is important to note that there is a significant body of research on first-order approximations of bilevel problems, such as the work by [R1]. Given the extensive nature of this field, we have opted to defer an in-depth exploration of first-order approximation methods to future investigations. This decision allows us to focus on the core contribution of this paper, which is the development of a novel data condensation framework for hyperparameter search.
> >     > [R1] Liu, Bo, Mao Ye, Stephen Wright, Peter Stone, and Qiang Liu. "Bome! bilevel optimization made easy: A simple first-order approach." Advances in Neural Information Processing Systems 35 (2022): 17248-17262.
> >
> >
> > > Weakness 3: I am curious about the details of the method to deal with discrete hyperparameters. The descriptions in Sec. 5.2 and Appendix H are confusing to me. An algorithmic flow is encouraged for better presentation. In addition, are other approaches like Gumbel Softmax applicable?
> >
> > **Response to Weakness 3:**  The details of dealing with discrete $\Lambda$, due to the space constraint, are deferred to Appendix G and H.
> >
> > - Specifically, we transform this problem as a continuous HPO, by defining an “interpolated” model (i.e., a supernet (Wang et al., 2020)) $f_\theta^\lambda$, where hyperparameters $\lambda = [\lambda^{(1)}, \ldots, \lambda^{(p)}] \in \Lambda$ and $\theta$ is the model parameter.
> > - This technique is known as the differentiable NAS approach (see Appendix B), e.g., DARTS (Liu et al., 2018), which usually follows a cell-based search space (Zoph et al., 2018) and continuously relaxes the original discrete search space. Subsequently, in (Xie et al., 2018; Dong & Yang, 2019), the Gumbel-softmax trick (Jang et al., 2017; Maddison et al., 2017) is applied to approximate the hyperparameter gradients. We apply the optimization strategy in GDAS (Dong & Yang, 2019), which also provides the approximations of the hypergradients in our experiments.
> >
> > > **Weakness 4:** Eq. IFT is an efficient approximation of the hypergradients. It would be better to make this clear in the description. And "=" should not be used in Eq. IFT.
> >
> > **Response to Weakness 4:** We will fix this in the revised manuscript, thank you for the suggestion.
> >
> >
> > > Weakness 5: Minor: some confusing expersions:
> > > - "this limitation" in the second paragraph of Introduction: the contents in the previous paragraph is not actually a limitation. And this sentence is incoherent to the last sentence of the previous paragraph.
> > > - The last two lines in Page 5.
> > > - Page 6: "For generating $S^{train}$, We ...", "W" here should be lowercase.
> >
> > **Response to Weakness 5**: We have incoporated those suggestions. Thank you!
> >
> > ---
> >
> > Again, we express our sincere gratitude to reviewer EsH1 for their valuable service in reviewing our manuscript.

---

### Official Review · Reviewer_sKjo · 2023-11-01

**Soundness:** 3 good
**Presentation:** 3 good
**Contribution:** 3 good
**Rating:** 6
**Confidence:** 3

**Summary:**

The paper presents a different view of data condensation, which is more focused on "hyperparameter search" across multiple architectures. It extends the original data condensation problems into hyperparameter-calibrated data condensation, which has several challenges: nested optimization and continuous hyperparameter search space. The proposed solution works very well compared to prior SOTA methodologies, showing a much higher rank correlation in Table 2. Also, the paper has some examples of its general use cases on image and graph domains.

**Strengths:**

1. The proposed problem makes sense and is important - this is an initial work focusing on hyperparameter search space in data condensation.
2. The solution shows high-performance improvement in terms of the rank correlation on several domains (image and graph)
3. The paper has a good balance between theoretical analysis and empirical understanding.

**Weaknesses:**

1. For the image domain, the used datasets are too simple, which has 32x32 pixels and a smaller number of classes. Usually, the hyperparameter search benefit is much higher in large resolution and large class datasets.

2. The current rank correlation metric is reasonable, but it would be extended to a more fine-granular level. For example, we can calculate the rank correlation for each class and aggregate them on average. This will show how the rank correlation matches with that obtained with the original dataset at the lower granularity.

**Questions:**

Refer to the weakness section.

**Details Of Ethics Concerns:**

I cannot find any ethical concern.

---

> ### Author Response · Authors · 2023-11-22
> **Response to Reviewer sKjo**
>
> We thank reviewer sKjo, for their service and constructive feedback. We appreciate that the reviewer recognizes the importance of the topic and the strengths of the paper. In particular, we are encouraged that reviewer sKjo finds that our proposed problem is important and novel, our solution is effective ('high-performance improvement'), our experiments are comprehensive and extensive on both image and graph data, and the paper has a good balance between theoretical analysis and empirical understanding.
>
> Below we address the concerns from reviewer sKjo.
>
> ---
>
> > **Weakness 1:** For the image domain, the used datasets are too simple, which has 32x32 pixels and a smaller number of classes. Usually, the hyperparameter search benefit is much higher in large resolution and large class datasets.
>
> **Response to Weakness 1:**
>
> - Our experimental focus of the paper is on the CIFAR-10 and CIFAR-100 datasets. We acknowledge the significance of testing on larger datasets like ImageNet. However, conducting Neural Architecture Search (NAS) experiments on the scale of ImageNet would be extremely time-consuming, making it hard to conduct in an academic setup. The efficacy and scalability of our dataset condensation approach are demonstrated through our successful application on CIFAR-10 and CIFAR-100, as well as large scale graph data. The main goal of our paper is to introduce an innovative approach to the community, as opposed to setting state-of-the-art NAS results on vast image classification datasets.
>
> - We thank the reviewer for this suggestion. We are in the process of evaluating our method on a large architecture. However, we would like to emphasize that
>     1. We have conducted architecture search over randomly sampled 100 network models from 15,625 networks for CIFAR-10 and CIFAR-100. Our experiments compare with 7 baselines. All experiments are repeated over 5 random seeds. So overall, we have worked over # of NETWORKS $\times$ (# of baselines +1) $\times$ random seeds = 100 $\times$ 8 $\times$ 5 = 4000 models. We believe this is a strong enough evidence that our methods outperform other with substantial statistical significance.
>     2. We have not only verifed the performance on vision tasks, but also have included hyperparameter search over Graph Neural Networks. Experimentally, we have shown 4 graph datasets including two large graphs with more than 100K nodes.
>     3. Our method is combined with off-the-shelf NAS algorithms. 100 models are used here for this exepriments as well.
>
> - A recent benchmarking study, "DC-BENCH: Dataset Condensation Benchmark," published at NeurIPS 2022 (https://arxiv.org/pdf/2207.09639.pdf), conducted experiments on CIFAR 10 and CIFAR 100 too. Consistent with this, our paper also includes experiments on CIFAR 10 and CIFAR 100, but additionally, we expand our investigation to include large-scale graph data.
>
> - We plan to address this experiment in future work, aiming to significantly enhance the method's efficiency and scalability. Despite these challenges, we believe our work merits publication due to its innovative approach and being the first to tackle the issue of hyperparameter search via data condensation.
>
>
> > **Weakness 2:** The current rank correlation metric is reasonable, but it would be extended to a more fine-granular level. For example, we can calculate the rank correlation for each class and aggregate them on average. This will show how the rank correlation matches with that obtained with the original dataset at the lower granularity
>
> Response to Weakness 2: We appreciate the suggestion to refine our rank correlation metric. Extending it to a more granular level, such as calculating the rank correlation for each class and then aggregating them on average, is indeed a valuable approach. This would provide a more detailed insight into how our rank correlation aligns with that obtained using the original dataset, particularly at a finer granularity. We will consider incorporating this enhanced metric in our future work to further substantiate our findings.
>
> ---
>
> Again, we express our sincere gratitude to reviewer sKjo for their valuable service in reviewing our manuscript.

---

### Official Review · Reviewer_4Foe · 2023-11-01

**Soundness:** 3 good
**Presentation:** 3 good
**Contribution:** 3 good
**Rating:** 5
**Confidence:** 4

**Summary:**

This paper proposes a novel hyperparameter-calibrated dataset condensation (HCDC) algorithm, to solve the problem of existing dataset distillation methods being difficult to migrate to models other than predefined ones. The validation set is condensed through hyperparameter gradient matching. The proposed method retains the ranking information on different hyperparameters and accelerates the search for hyperparameter and architecture.

**Strengths:**

1. The hyperparameter gradient matching proposed in the paper is relatively novel and provides new ideas for improving the generalization performance of dataset distillation methods;
2. The paper provides a solid theoretical analysis and a clear explanation of the research issues;
3. The proposed method achieves good results on image datasets and graph datasets.

**Weaknesses:**

1. The paper does not explain some concepts mentioned for the first time, such as "supernet";
2. Lack of some ablation experiments, such as the performance of searching directly by compressing the test set using the dataset distillation method;
3. As a dataset distillation method, there is a lack of comparison with other dataset distillation methods in terms of compression rate and generalization performance.

**Questions:**

1. In image dataset distillation, is it possible for methods to be applicable to distillation models of different architectures? For example ConvNet, VGG, ResNet, etc.
2. What is the difference between distilling directly on the test set and then using it for hyperparameter search compared to using distilled samples on a randomly selected training set?
3. Do different sampling methods have an impact when sampling from pre-distilled samples?
4. What is the time cost of the method? Is it possible to extend from validation set compression to training set compression?

---

> ### Author Response · Authors · 2023-11-22
> **Response to Reviewer 4Foe (1/2)**
>
> We appreciate the reviews provided by reviewer 4Foe. We are gratified by the recognition of the paper's key strengths. In particular, we are pleased that reviewer 4Foe acknowledges the innovation of our hyperparameter gradient matching method, the solidness of our theoretical analysis, the clarity in explaining research issues, and the impressive results achieved with image and graph datasets.
>
> Below, we will address all questions and concerns raised by reviewer 4Foe, which primarily stem from misunderstandings.
>
>
> > **Weakness 1:** The paper does not explain some concepts mentioned for the first time, such as "supernet".
>
> **Response to Weakness 1:** Thank you for your valuable feedback regarding the accessibility of our paper. In the main manuscript, we deliberately chose not to delve into the concept of 'supernet' to avoid potential confusion, considering it is not central to the paper's primary message. We did mention it briefly in the conclusion section. The term 'supernet' is referenced twice in the appendices: once in Appendix D and again in Appendix G. We acknowledge that a comprehensive explanation and citation for 'supernet' were provided only in Appendix G. Following the reviewer's suggestion, we have now updated the manuscript to not mention the term 'supernet' in the main paper, to ensure consistency and clarity.
>
> ---
>
> > **Weakness 2:** Lack of some ablation experiments, such as the performance of searching directly by compressing the test set using the dataset distillation method.
>
> **Response to Weakness 2:** Regarding Weakness 2, there seems to be a misconception.
>
> - Tables 1 and 2 in our paper actually present the experiments the reviewer is referring to. As our work is the **first** in integrating hyperparameter calibration with dataset condensation, our comparison is with the current methods that directly compress the validation set. (We infer the reviewer's mention of the 'test set' to be a reference to the 'validation set,' since the test set is not typically accessible during training and hyperparameter tuning.)
>
> - The results in Tables 1 and 2 demonstrate HCDC's remarkable ability to preserve performance across varying hyperparameters. This underscores HCDC's efficacy in condensing the dataset to speedup the hyperparameter search process.
>
> ---
>
> > **Weakness 3:** As a dataset distillation method, there is a lack of comparison with other dataset distillation methods in terms of compression rate and generalization performance.
>
> **Response to Weakness 3:** There appears to be a misunderstanding regarding the goal of the paper, the novelty and comparison scope of our method.
>
> - Our approach is the first of its kind focusing on data condensation specifically for hyperparameter search, filling a gap in existing methodologies.
>
> - We have conducted comparative analyses with other data distillation techniques, as showcased in Tables 1 and 2. Table 1 focuses on image data, where we standardized the compression to 50 images per class for all baselines. This was to maintain consistency, as varying the compression rate was not the primary focus of our image data experiments. In contrast, Table 2, which centers on graph data, includes comparisons with state-of-the-art graph compressing methods under different compression rates ($c_{train}/n$).
>
>
> - It is essential to highlight that our objective extends beyond simple data compression. **The primary aim is to leverage data condensation to speedup hyperparameter search, which is often the most resource/computation-intensive aspect of machine learning model development.** Once we have successfully identified the hyperparameters, it's not necessary to train the final model on condensed data. We can instead train the model on the full dataset using the identified hyperparameters.

---

> > ### Author Response · Authors · 2023-11-22
> > **Response to Reviewer 4Foe (2/2)**
> >
> > > **Question 1:** In image dataset distillation, is it possible for methods to be applicable to distillation models of different architectures? For example ConvNet, VGG, ResNet, etc.
> >
> > **Response to Question 1:** Yes, definitely. The strength of our method lies in its ability to preserve performance across various models with different architectures and hyperparameters, such as ConvNet, VGG, and ResNet.
> >
> > > **Question 2:** What is the difference between distilling directly on the test set and then using it for hyperparameter search compared to using distilled samples on a randomly selected training set?
> >
> > **Response to Question 2:** As shown in our experimental results, the latter will not perserve performance across different hyperparameters. So hyperparameter search on those condensed dataset will fail, i.e., hyperparameter A on the condensed dataset perform better than hyperparameter B, but on the original full dataset, hyperparameter A will perform worse than hyperparameter B.
> >
> > > **Question 3:** Do different sampling methods have an impact when sampling from pre-distilled samples?
> >
> > **Response to Question 3:** We may have misunderstood your question, but it seems you're inquiring about the mini-batch sampling method we applied to the distilled dataset. In our approach, mini-batch sampling is typically unnecessary due to the small size of the distilled dataset. Additionally, we don't anticipate significant differences in outcomes with other sampling methods, such as random sampling.
> >
> > > **Question 4:** What is the time cost of the method? Is it possible to extend from validation set compression to training set compression?
> >
> > **Response to Question 4:** We believe this question is due to a misunderstand of our method, and our goal. After clarification above, hopefully we have resolved the confusion.
> >
> > ---
> >
> > Again, we express our sincere gratitude to reviewer 4Foe for their valuable service in reviewing our manuscript.

---

### Official Review · Reviewer_UefQ · 2023-11-09

**Soundness:** 2 fair
**Presentation:** 2 fair
**Contribution:** 3 good
**Rating:** 3
**Confidence:** 4

**Summary:**

This paper introduces a novel method for optimizing a synthetic validation set to align the model performance rankings between the condensed and original datasets. To achieve this, the authors introduce a hypergradient alignment objective function and devise an algorithm that employs implicit differentiation and inverse Hessian approximation. Through a comprehensive series of experiments conducted on datasets in both image and graph domains, the authors effectively demonstrate how their optimized synthetic dataset enhances the results of architecture and hyperparameter search when using the condensed dataset.

**Strengths:**

- The paper introduces a novel approach to condensing datasets for architecture and hyperparameter search.
- This paper effectively tackles the important challenges within the field of dataset condensation research.
- The primary motivation behind the main objective is compelling.
- The proposed method shows strong performance in architecture and hyperparameter search across various datasets.

**Weaknesses:**

- Some technical sections of the paper are hard to understand.
  - On page 5, Definition 2: Is "cos" representing cosine-similarity? If so, why should the term between two hypergradients be zero?
  - In section 5.2 on page 6, how can discrete factors like model depth or kernel size be expanded continuously?
- The process of condensing synthetic validation appears intricate and time-intensive. However, the paper lacks a comprehensive analysis of this procedure.
  - Could you provide information on the time required to optimize the dataset using the HCDC algorithm in experiments, including Table 3?
- Figure 4 doesn't offer a fair comparison of the architecture search. To achieve this, the graph should be revised to include the optimization time for the HCDC algorithm.

typo
- p.2: $\mathcal{T}$ with $\mathcal{T}^{\text{train}}$ -> $\mathcal{T}^{\text{train}}$ with $\mathcal{T}$
- p.6: , We employ -> , we employ
- p.6: ).Specifically -> ). Specifically

**Questions:**

See the weakness section above.

---

> ### Author Response · Authors · 2023-11-22
> **Response to Reviewer UefQ (1/2)**
>
> We express our sincere gratitude to reviewer UefQ for their insightful review and constructive feedback. We are particularly encouraged by reviewer UefQ's recognition of the novelty of our approach to dataset condensation for architecture and hyperparameter search, its effectiveness in addressing significant challenges within the field, the compelling rationale behind our primary objective, and the impressive performance demonstrated by our proposed method across various datasets.
>
>
> Reviewer UefQ's primary concern centers on the lack of a comprehensive analysis of the time complexity of our proposed method. We emphasize that this **one-time cost** grants significant computation and time savings for **repeated hyperparameter search cost**.
>
> Considering all the other merits of the paper, we sincerely hope that Reviewer UefQ could consider adjusting their score after our more detailed clarification below.
>
> ---
>
> > **Weakness 1:** The process of condensing synthetic validation appears intricate and time-intensive. However, the paper lacks a comprehensive analysis of this procedure.
> Could you provide information on the time required to optimize the dataset using the HCDC algorithm in experiments, including Table 3?
>
> **Response to Weakness 1:**
>
> - First, the primary motivation behind our Hyperparameter-Calibrated Dataset Condensation (HCDC) approach is to expedite the hyperparameter search process when dealing with extensive validation datasets. While the condensation process introduces an overhead, the subsequent hyperparameter search on the condensed dataset is considerably faster. **Our empirical evaluations (*Table 3*) demonstrate that the overall time savings achieved by our method outweigh the initial condensation overhead**, making it a viable alternative to traditional hyperparameter search on the original dataset.
>
> - It's important to note that the dataset condensation process is a one-time cost. Once we have the condensed dataset, it serves as a tool to significantly expedite numerous hyperparameter/architecture search algorithms. Our HCDC considerably expedite the two NAS baselines (DARTS-PT and REINFORCE) as shown in **Table 3**.
>
> - The HCDC condensation process requires a one-time investment of approximately 127.9 minutes. Although this duration seems substantial, the time savings in subsequent architecture searches are considerable. Specifically, the search time for DARTS-PT is reduced from 229 seconds to 35.5 seconds and for REINFORCE from 1492 seconds to 119 seconds, as detailed in Table 3. This significant relative reduction in search times justifies the initial longer condensation period, as it provides substantial efficiencies for any future searches conducted on the condensed dataset.
>
> - Regarding the time complexity of HCDC condensation (Algorithm 1), we have conducted a theoretical complexity analysis presented in Appendix I. In summary, while the conventional gradient-matching-based dataset condensation has a per iteration time complexity of $O(P)$, where $P$ is the number of model parameters, the complexity of HCDC is $O(P+H)$, with $H$ being the number of hyperparameters. Owing to the efficient inverse Hessian approximation, the linear complexity concerning both parameters and hyperparameters is maintained. As evidence, our "Finding the best convolution filter on graphs" experiment demonstrated that, with a few introduced hyperparameters, the HCDC runtime on the ogbn-arxiv graph dataset (with a condensation ratio $c_{train}/n$ of 0.5%) was 805s, compared to the standard condensation's 494s. We intend to include more empirical runtime data for HCDC in the revised manuscript, as you suggested.
>
> ---
>
> > **Weakness 2:** Figure 4 doesn't offer a fair comparison of the architecture search. To achieve this, the graph should be revised to include the optimization time for the HCDC algorithm.
>
> **Response to Weakness 2: **
> - As mentioned in the previous response, the one-time running time needed for HCDC is around 127.9 minutes. It's important to note that the dataset condensation process is a one-time cost. Once we have the condensed dataset, it serves as a tool to significantly expedite numerous hyperparameter/architecture search algorithms.

---

> > ### Author Response · Authors · 2023-11-22
> > **Response to Reviewer UefQ (2/2)**
> >
> > > **Weakness 3:** Some technical sections of the paper are hard to understand.
> >     > - On page 5, Definition 2: Is "cos" representing cosine-similarity? If so, why should the term between two hypergradients be zero?
> >     > - In section 5.2 on page 6, how can discrete factors like model depth or kernel size be expanded continuously?
> >
> > **Response to Weakness 3: **
> > Thanks for the detailed feedback. The confusion is due to lack of clarity, and we have fixed our manuscript to clarify them.
> > - In our manuscript, "cos" represents cosine distance. According to Wikipedia page here https://en.wikipedia.org/wiki/Cosine_similarity, cosine distance is commonly used for the complement of cosine similarity in positive space, that is cosine distance = 1 - cosine similarity. So we minimize the cosine distance to maximize the cosine similarity. We will clarify this in the revised manuscript.
> >
> >
> > - The details of dealing with discrete $\Lambda$, due to the space constraint, are deferred to Appendix G and H.
> >     - Specifically, we transform this problem as a continuous HPO, by defining an “interpolated” model (i.e., a supernet (Wang et al., 2020)) $f_\theta^\lambda$, where hyperparameters $\lambda = [\lambda^{(1)}, \ldots, \lambda^{(p)}] \in \Lambda$ and $\theta$ is the model parameter.
> >     - This technique is known as the differentiable NAS approach (see Appendix B), e.g., DARTS (Liu et al., 2018), which usually follows a cell-based search space (Zoph et al., 2018) and continuously relaxes the original discrete search space. Subsequently, in (Xie et al., 2018; Dong & Yang, 2019), the Gumbel-softmax trick (Jang et al., 2017; Maddison et al., 2017) is applied to approximate the hyperparameter gradients. We apply the optimization strategy in GDAS (Dong & Yang, 2019), which also provides the approximations of the hypergradients in our experiments.
> >
> > ---
> >
> > Again, we express our sincere gratitude to reviewer UefQ for their valuable service in reviewing our manuscript.

---

### Author Response · Authors · 2023-11-22
**Global Response to Reviewers**

We are grateful for the constructive feedback from reviewers UefQ, sKjo, EsH1, and 4Foe. We appreciate their general acknowledgment of the motivation, methodology, and effectiveness of our HCDC approach, as well as their suggestions for improvement. We have carefully incoporated their feedback and made improvements to our work accordingly.

Our paper received positive feedback for its strengths from the reviewers. Reviewer UefQ acknowledged the novelty of our approach to dataset condensation for architecture and hyperparameter search, its effectiveness in significant field challenges, the compelling rationale behind our primary objective, and the impressive performance across various datasets. Reviewer sKjo recognized the importance and novelty of our proposed problem, the effectiveness of our solution, the comprehensiveness of our experiments on image and graph data, and the good balance between theoretical analysis and empirical understanding. Reviewer EsH1 appreciated the importance and novelty of our proposed problem, the quality and coherence of our presentation, the significance of our demonstration of existing methods' drawbacks, and the well-motivated, reasonable, and interesting nature of our method. Lastly, reviewer 4Foe commended the innovation of our hyperparameter gradient matching method, the solidness of our theoretical analysis, the clarity in research issue explanation, and the impressive results achieved with image and graph datasets.

We have addressed all weaknesses and questions raised in each individual response. We have incorporated the suggested changes and submitted the revised manuscript. Please let us know if you have any further questions.

Again, we extend our sincere gratitude to all reviewers for their valuable time and effort.

---

### Meta-Review · Area_Chair_HRqy · 2023-11-30

**Metareview:**

The reviewers and meta reviewer all carefully checked and discussed the rebuttal. They thank the author(s) for their response.

While the reviewers praised the relevance and originality of the tackled problem—i.e., dataset condensation preserving the ordering of hyperparameter rankings with the goal of speeding up hyperparameter tuning—the submission still suffers from several weaknesses and limitations, which all put together warrants further investigations and consolidations.

For example:

* Clarity of the manuscript as well as its self-containedness. For instance, while the optimisation w.r.t. S (see Eq. 2) is a key component of the method (line 8 in Algorithm 1), we cannot understand it from the only core paper how it is done.

* One major issue with the submission (related to the point above) is that it tries to cover too many aspects without being able to investigate them with sufficient depth. For instance, the only continuous relaxation used here would deserve multiple ablation studies to support all design choices. The same goes for the impact of the IFT approximation and of the linear Taylor approximation (in Sec. 4).

* In the Bayesian optimization literature, there are many approaches (e.g., BOHB, hyperband) that help speed up hyperparameter tuning by using subsampling and/or early feedback (e.g., decisions after a few steps). Such baselines would be important to add since they have the same objective. Comparing favorably to those would help demonstrate the value of the dataset condensation approach.

* The proof of Th 1 (see Appendix F.3) seems problematic. When $\lambda$ is a vector, the multivariate mean-value theorem needs to be used while, right now, everything is derived as if $\nabla_\lambda L$ was a scalar. Moreover, while the necessity condition seems clear, the other direction is not. In 2-D, it is easy to find a contradiction: with $a=[-1, -1], b=[-2, 0.5], d = [1, 2]$, we have $(a^\top d) \cdot (b^\top d) = 3 > 0$ but $1 - \cos(a,b) \approx 0.48 \neq 0$.
If this is the case, the link between Definition 1 and Definition 2 should be revised and the choice of the objective function in Eq. HCDC re-discussed.

* Finally, while the reviewers and meta reviewer do acknowledge the breadth of the already conducted experiments, investigating the behavior of the approach with an increasing scale would be important.


All in all, the paper is recommended for a rejection. We are convinced that the suggestions surfaced through the reviews will help strengthen the paper for a future resubmission, which the reviewers and meta reviewer all encourage.

**Justification For Why Not Higher Score:**

* Clarity & self-containedness of the manuscript
* Density/organisation of the manuscript (and the appendix---with a lot of unreviewed material), which prevents from studying in depth several key components of the proposed method
* Missing relevant baselines aiming at solving the same problem
* In its current form, the paper fails to motivate why the complexity of the approach is required. Designing simpler baselines would help highlight the challenges at hand. For example, could we solve (2) where we would find a single S that simultaneously minimise the objective function  averaged over $M$ models indexed by $M$ different hyperparameters? ($\min_S \sum_{m=1}^M ...\text{loss function of Eq. 2 with }\theta_m....$) In that case, this simple generalisation of SDC may already (or not) work better over different hyperparameters.

**Justification For Why Not Lower Score:**

N/A

---

### Decision · Program_Chairs · 2024-01-16

Reject